# SAR: Scene-Action Representation for End-to-End Autonomous Driving

## Abstract

End-to-end autonomous driving systems have made remarkable progress by integrating perception, prediction, and planning into a fully differentiable framework. However, most existing methods either rely heavily on dense intermediate supervision (e.g., segmentation and mapping) or neglect behavior modeling, which leads to significant trajectory deviations and safety risks in highly interactive scenarios. To address these challenges, we propose a novel end-to-end Scene Action Representation (*SAR*) framework that enhances sparse scene modeling through structured behavior injection. Inspired by human driving cognition, *SAR* decomposes the scene into three complementary components: sparse scene semantics, ego-action awareness, and multi-agent action awareness. These components are fused via a specially designed *Scene-Action Transformer* to produce a consistent, interpretable and interaction-aware representation for high-quality trajectory planning. Unlike prior approaches, *SAR* achieves strong generalization in highly interactive urban scenarios with only a small annotation cost. Experimental results on the nuScenes benchmark show that SAR reduces L2 trajectory error by 47% and collision rate by 41% compared to VAD. It also demonstrates superior robustness on NAVSIM and Bench2Drive, achieving new state-of-the-art performance in both open-loop and closed-loop evaluations. The code will be released soon.

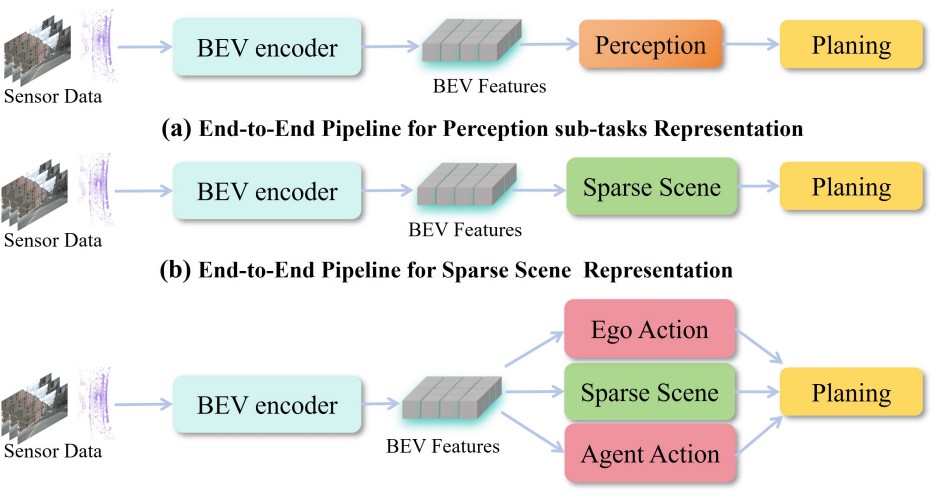

(a) End-to-End Pipeline for Perception sub-tasks Representation

(b) End-to-End Pipeline for Sparse Scene Representation

(c) End-to-End Pipeline for Scene-Action Representation(Our)

Figure 1: **Different architectures for scene representation.** (a) Scene representation constructed via multiple perception sub-tasks. (b) Purely token-based sparse scene representation. (c) Our proposed sparse scene–action representation.

# 1 INTRODUCTION

With the rapid development of autonomous driving technology, achieving accurate and robust understanding of complex and dynamic traffic environments has become a main challenge. Traditional systems typically adopt a modular pipeline architecture, where perception, prediction, planning, and control are separated into distinct stages. Although this design offers advantages in terms of engineering flexibility and interpretability, it is prone to error propagation and lacks global optimization, particularly in multi-agent and highly interactive scenarios.

In recent years, an end-to-end paradigm has emerged to directly generate driving trajectories from raw sensor data through fully differentiable models. This paradigm simplifies the network architecture and reduces information loss across stages. For example, UniAD Hu et al. (2023), VAD Jiang et al. (2023), and GenAD Zheng et al. (2024b) integrate perception, prediction, and planning into an unified framework, achieving notable improvements in coordination and performance. However, these methods still heavily rely on dense Bird's-Eye-View (BEV) representations and intermediate supervision signals (e.g., detection, tracking, and segmentation), leading to high annotation costs and computational overhead, while exhibiting limited generalization in long-tail or interaction-intensive scenarios.

To mitigate the dependence on dense features, recent works have explored sparse scene modeling, which represents the environment with a small number of semantic tokens Li & Cui (2025) or queries Sun et al. (2024), significantly reducing computational cost. However, sparse scene representations often lack of sufficient dynamic behavioral information and fine-grained semantics, making them inadequate for complex real-world traffic scenarios, particularly in crowded intersections, severe occlusions, or highly interactive situations, leading to poor planning performance. This raises a key research question: ***How can we develop a behavior-aware modeling approach to guide and enhance sparse scene representations for reliable trajectory planning?***

To enhance the expressiveness of sparse scene representations in complex traffic environments, we propose an end-to-end **Scene-Action Representation (SAR)** framework for trajectory prediction in autonomous driving, as shown in Figure 1. Different from previous end-to-end approaches, SAR defines the global driving environment representation as a joint embedding of sparse scene semantics and ego/agent actions. Specifically, SAR consists of three complementary components: an environment component, an ego component, and an agent component, which model the global scene, ego action awareness, and multi-agent action awareness,respectively. Then, SAR introduces a **Scene–Action Transformer** to capture cross-modal interactions between scene context and ego–agent actions for unified trajectory reasoning, enabling accurate and robust trajectory generation. By tightly coupling semantics and behaviors, SAR significantly improves the modeling capacity of sparse representations in highly interactive multi-agent scenarios.

To the best of our knowledge, SAR is the first framework that leverages ego–agent actions to guide sparse scene queries for driving environment representation. Compared to existing approaches Chen et al. (2024); Jia et al. (2025); Lin et al. (2022), SAR significantly reduces the reliance on intermediate supervision and computational cost, requiring only agent positions and trajectory supervision to achieve optimal performance.

Our main contributions are summarized as follows:

- We propose an end-to-end Scene Action Representation framework and introduce a ***Scene–Action Transformer***, which enhances sparse scene modeling by incorporating ego and multi-agent action awareness.

- We design a risk aware ***Ego Action Encoder*** which can effectively guide scene queries to focus on ego-centric risk features.

- Experiments on three challenging benchmarks: *nuScenes* Caesar et al. (2020) (open-loop trajectory prediction), *NAVSIM* Dauner et al. (2024) (perception-driven control), and *Bench2Drive* Jia et al. (2024) (long-horizon closed-loop simulation)—demonstrate that SAR achieves state-of-the-art performance in both open-loop and closed-loop settings, showcasing its efficiency and scalability in real-world autonomous driving tasks.

## 2 RELATED WORK

### 2.1 END-TO-END AUTONOMOUS DRIVING

End-to-end autonomous driving (E2E-AD) aims to directly map raw sensor inputs to control commands or trajectories via fully differentiable models, avoiding the modular decomposition of perception, prediction, and planning Bojarski et al. (2016); Dosovitskiy et al. (2017); Chen et al. (2020); Cao et al. (2025). This unified design helps mitigate error propagation and inter-module instability.

Early methods like CIL Codevilla et al. (2018) and CILRS Codevilla et al. (2019) used CNNs to regress control signals from monocular images, with auxiliary velocity prediction. LBC Chen et al. (2020) introduced a privileged expert model to guide policy learning, while Hydra-MDP Li et al. (2024b) combined affordance learning and knowledge distillation from both rule-based and human demonstrations.

Recent works incorporate multi-modal sensing and Transformer-based fusion. Transfuser Chitta et al. (2022) and Interfuser Shao et al. (2023a) fuse camera and LiDAR with rule-based priors. MMFN Zhang et al. (2022) uses VectorNet Liu et al. (2022) for map encoding, ThinkTwice Jia et al. (2023) adopts a DETR-style decoder Wang et al. (2022), and ReasonNet Shao et al. (2023b) enhances temporal reasoning.

Despite progress, many E2E-AD systems still struggle with interpretability and multi-agent reasoning in complex scenes.

### 2.2 SCENE REPRESENTATION IN END-TO-END AUTONOMOUS DRIVING

Scene representation is key for connecting perception to decision-making. A common strategy is to construct intermediate representations typically in Bird's-Eye View (BEV) from multi-modal inputs. ST-P3 Hu et al. (2022) jointly trains perception, prediction, and planning via BEV segmentation. UniAD Hu et al. (2023) extends this with a full Transformer stack and adds occupancy prediction, 3D detection, and tracking.

To reduce supervision cost, vectorized representations have gained popularity. VAD Jiang et al. (2023) and VADv2 Chen et al. (2024) use vector-based BEV objects to simplify annotations while retaining semantic clarity. PARA-Drive Weng et al. (2024) restructures pipelines into parallel tasks. OccWorld Zheng et al. (2024a) introduces a 3D world model for dynamic scene evolution. Drive-Transformer Jia et al. (2025) unifies all tasks under a parallel Transformer.

However, dense BEV grids remain costly. Recent sparse methods address this by using instance-centric features. SparseDrive Sun et al. (2024), DiffusionDrive Liao et al. (2024) and SparseAD Zhang et al. (2024) decouple perception into sparse tokens. GaussianFusion Liu et al. (2025) compresses inputs with global Gaussian tokens. SSR Li & Cui (2025), LAW Li et al. (2024a), and World4Drive Zheng et al. (2025) eliminate intermediate labels via latent, self-supervised world models.

Yet most sparse methods focus on perception abstraction, overlooking behavior modeling, especially in long-tail scenarios. To address this, our SAR framework introduces structured agent-scene interactions and action-intent modeling, enhancing decision robustness in complex environments.

## 3 METHOD

### 3.1 OVERVIEW

Fig. 2 presents an overview of our SAR framework, which consists of three core components: *Sparse Scene Tokenization*, *Ego Action Decoder* and *Scene–Action Transformer*. *Sparse Scene Tokenization* extracts compact scene representations from multi-modal BEV features using a tokenization mechanism. *Ego Action Decoder* integrates the BEV context, agent positions and navigation commands to model the collision risks between the ego and surrounding agents, and *Scene–Action Transformer* captures cross-modal interactions between scene context and ego-agent dynamics for unified trajectory reasoning.

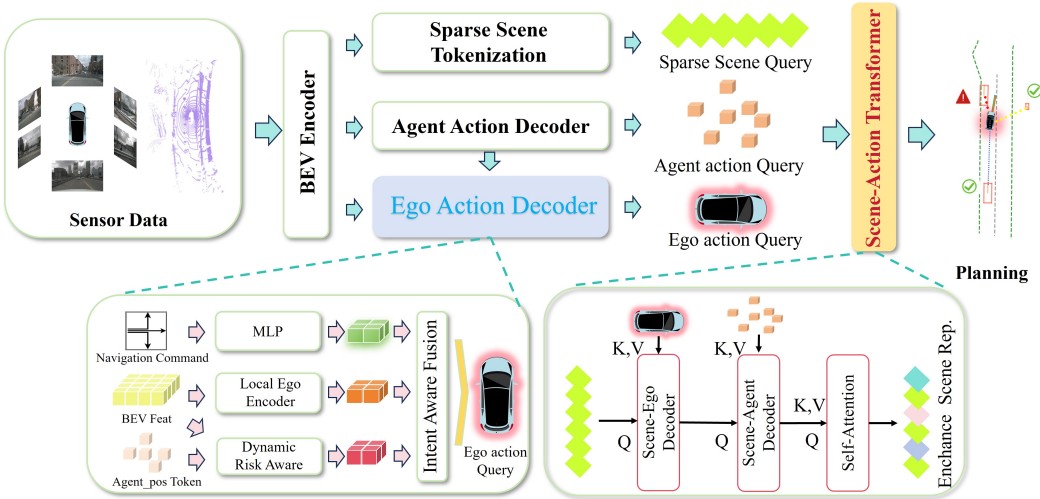

Figure 2: **The framework of our Scene-Action Representation for the end-to-end autonomous driving.** SAR uses multi-sensor data as input, leveraging BEV features converted into sparse scene query ,agent action query and ego action query. A Scene-Action Transformer module is used to enhance scene representation through cascaded cross-modal interactions, which is essential for trajectory prediction.

## 3.2 SPARSE SCENE TOKENIZATION

We construct an unified multi-modal BEV feature map by combining LiDAR and multi-view image features by following the general paradigm of BEVFormer Li et al. (2022). Meanwhile, temporal information is injected via a cross-attention with the previous BEV, and spatial semantics is enriched by a deformable attention to image features:

$$\mathcal{B}_t = DefAttn\big(CrossAttn(\mathcal{B}_{lidar}, \mathcal{B}_{t-1}), \mathcal{F}_{img}\big). \tag{1}$$

where $\mathcal{B}_t$ is the unified BEV feature map at time step $t$. $\mathcal{B}_{lidar}$ and $\mathcal{B}_{t-1}$ are the BEV feature maps from LiDAR and the previous time step, respectively. $\mathcal{F}_{img}$ represents the features extracted from multi-view images. $CrossAttn$ and $DefAttn$ are the cross-attention and deformable attention mechanisms, respectively.

To obtain a compact yet informative representation, we follow the sparse scene representation approach in Li & Cui (2025). The BEV features are flattened and then fused with a navigation-conditioned embedding through a lightweight gating mechanism, and further passed into a learnable token extractor that selects $N_s$ informative scene tokens:

$$\mathcal{Q}_{scene} = TokenLearner([\mathcal{B}_{seq}|\mathcal{P}_{seq}]). \tag{2}$$

The resulting content and positional embeddings $\mathcal{Q}_{scene}, \mathcal{P}_{scene}$ serve as input to the subsequent Scene-Action Transformer for joint scene understanding and trajectory planning. Here, $\mathcal{Q}_{scene}$ is the set of $N_s$ selected scene tokens. $\mathcal{B}_{seq}$ and $\mathcal{P}_{seq}$ are the flattened BEV feature sequence and the navigation conditioned positional embedding. The $[\cdot|\cdot]$ denotes a concatenation operation, and $TokenLearner$ is the learnable module for token extraction, more technical details are provided in the Appendix A.1.

## 3.3 EGO ACTION DECODER

While sparse scene representations can capture global context from dense BEV features, they inherently lack ego state awareness. To address this limitation, we introduce a **Ego Action Decoder**– a lightweight ego query decoding module that consists of three components: (i)**Local Ego Encoder**, which encodes semantic features of the immediate surroundings of the ego vehicle; (ii)**Dynamic Risk Awareness Modeling**, which models the ego vehicle's dynamic risk perception; and (iii)**Intent Aware Query Fusion**, which adapts ego queries to varying driving scenarios through intent-aware fusion.

**Local Ego Encoder** We first extract a region of interest (ROI) around the ego vehicle from the BEV feature space. Specifically, we apply a binary mask $M_{\text{center}}$ to the BEV feature map $B_t \in \mathbb{R}^{H \times W \times C}$ to retain only the local neighborhood. The masked features are then passed through a convolution layer followed by average pooling to obtain a compact local BEV feature $B_{\text{center}}$:

$$B_{\text{center}} = \text{AvgPool}\big(\text{Conv}_{\text{roi}}(B_t \odot M_{\text{center}})\big). \tag{3}$$

Here, $M_{\text{center}}$ is a binary mask, $B_t$ is the BEV feature map at the current time step, and $B_{\text{center}}$ is the resulting local BEV feature. Concretely, $M_{\text{center}}(i,j) = 1$ if the BEV cell $(i,j)$ lies inside a rectangular ROI $[-R_x, R_x] \times [-R_y, R_y]$ centered at the ego in the BEV frame, and $M_{\text{center}}(i,j) = 0$ otherwise. In practice, we choose $(R_x, R_y)$ such that the ROI covers roughly one quarter of the full BEV range (see Appendix A.5 for an ablation on this distance parameter). This encoder thus focuses on distilling the most relevant local semantic context for the ego vehicle from the global BEV features, providing a refined input for decision-making.

**Dynamic Risk Awareness Modeling** To model the potential collision risk posed by surrounding traffic participants, we propose a dynamic, physics-based risk assessment method that goes beyond simple distance-based metrics. Recognizing that collision risk critically depends on relative motion, we incorporate the key traffic domain concepts of Time-to-Collision (TTC) and Distance to Closest Point of Approach (DCPA).In Appendix A.4 we provide the theoretical definitions of TTC and DCPA.

Instead of introducing an extra supervised trajectory prediction module, we derive the quantities needed for TTC/DCPA directly from the *latent agent/ego states* produced by the shared backbone and the existing detection/planning heads. Concretely, the ego branch provides the planned ego motion, while the agent branch provides per-agent states inferred from detection features; no additional trajectory loss is imposed on agents. Based on these latent states, we compute TTC and DCPA for each ego–agent pair and construct a soft, collision-aware risk embedding $f_{\text{risk}}$ that combines distance and temporal factors:

$$f_{risk} = \mathbb{I}(DCPA < \delta \wedge TTC < \gamma) \cdot \Big( \frac{\lambda_d}{DCPA + \epsilon} + \frac{\lambda_t}{TTC + \epsilon} \Big). \tag{4}$$

Here, $\mathbb{I}(\cdot)$ is an indicator function, $\delta$ and $\gamma$ are learnable threshold, $\lambda_d$ and $\lambda_t$ are learnable weights. This representation ensures that a strong risk signal is generated only when a potential collision is both probable and imminent.

**Intent Aware Query Fusion** To dynamically fuse the spatial semantics ($B_{center}$), the ego risk awareness($f_{risk}$), and the ego vehicle's navigational goals ($f_{navi}$), we introduce a gated attention mechanism. We emphasize that the intent here refers to our own navigational objective, not the intentions of other agents. We concatenate these features and feed them into a learnable gating layer $W_{gate} \in \mathbb{R}^{3 \times 3D}$ to compute attention weights:

$$[\alpha_{bev}, \alpha_{risk}, \alpha_{navi}] = Softmax\left(W_{gate}[B_{center}; f_{risk}; f_{navi}]\right). \tag{5}$$

Where the weights $\alpha_{bev}$, $\alpha_{risk}$, and $\alpha_{navi}$ represent the dynamically learned importance of each information modality. For example, in dense traffic, higher weights are assigned to $\alpha_{bev}$ and $\alpha_{risk}$; whereas on an open road, $\alpha_{navi}$ is given more importance. The final ego-centric query $Q_{ego}$ is then computed as a weighted sum of the features:

$$Q_{ego} = \alpha_{bev} \cdot B_{center} + \alpha_{risk} \cdot f_{risk} + \alpha_{navi} \cdot f_{navi}. \tag{6}$$

The resulting query $Q_{ego} \in \mathbb{R}^D$ serves as a compact and rich representation that integrates fine-grained semantic context, precise positional information, and dynamic navigational intent, providing a solid foundation for decision-making.

## 3.4 AGENT ACTION DECODER.

At first, we initialize the agent queries $\mathcal{Q}_{agent}$ and extract motion-centric features from the LiDAR BEV representation $\mathcal{B}_{lidar}$ via a deformable attention:

$$\hat{\mathcal{Q}}_{agent} = DefAttn(\mathcal{Q}_{agent}, \mathcal{B}_{lidar}). \tag{7}$$

Notably, we only adopt the LiDAR BEV features rather than the fused multi-modal BEV features to focus the network's attention on precise motion cues for agent reasoning.

To model multi-modal future behavior, each agent query is augmented with $M$ learnable mode embeddings $\mathcal{E}_{mode}$:

$$\mathcal{Q}_{motion} = \hat{\mathcal{Q}}_{agent} + \mathcal{E}_{mode}. \tag{8}$$

To capture the dependencies among different behavior modes of the same agent, we apply a self-attention mechanism over the motion queries $\mathcal{Q}_{motion}$. This allows information exchange across modes, enabling the model to learn potential cooperative or exclusive relationships:

$$\hat{\mathcal{Q}}_{motion} = SelfAttn(\mathcal{Q}_{motion}). \tag{9}$$

We then predict a confidence score $s_{i,m}$ for each agent mode pair $(i, m)$ and select only the top-$K$ high-confidence motion queries:

$$\tilde{\mathcal{Q}}_{motion} = TopK(\hat{\mathcal{Q}}_{motion}, s), \tag{10}$$

which is used as $\mathcal{Q}_{motion}$ in the subsequent Scene–Agent interaction.

### 3.5 SCENE-ACTION TRANSFORMER

To construct an unified driving scene representation that effectively integrates ego vehicle intent and the dynamic behaviors of surrounding agents, we propose a ***Scene-Action Transformer***: a multi-stage reasoning mechanism that progressively injects motion-aware queries into sparse scene tokens via a cascade of attention-based operations. This mechanism includes three sequential stages: (i) **Scene-Ego Decoder** ego-to-scene intention injection, (ii) **Scene-Agent Decoder** agent-to-scene motion fusion, and (iii) **Latent Scene Self-refinement**.

**Scene-Ego Decoder.** To align the global scene representation with ego-centric planning intent, the ego motion query $\mathcal{Q}_{ego}$ produced by the risk-aware fusion module, is injected into the sparse scene queries via the cross-attention as follows:

$$\hat{\mathcal{Q}}_{scene} = CrossAttn(\mathcal{Q}_{scene}, \mathcal{Q}_{ego}, \mathcal{Q}_{ego}) \tag{11}$$

Spatial consistency is maintained through positional encoding. This step explicitly introduces anchors ego intent into the global context, enabling intention-driven spatial reasoning within the scene.

**Scene-Agent Decoder.** We further enhance the behavioral expressiveness of the scene representation by incorporating dynamic cues from surrounding agents. A confidence-based selection mechanism is applied to identify top-ranked agent motion queries, followed by cross-attention fusion:

$$\mathcal{R}_{scene} = CrossAttn(\hat{\mathcal{Q}}_{scene}, \hat{\mathcal{Q}}_{motion}, \hat{\mathcal{Q}}_{motion}) \tag{12}$$

where $\mathcal{R}_{scene}$ is the enhanced scene representation. This operation embeds the multi-agent behavioral interactions into the global scene representation, enriching the contextual understanding required for complex decision-making.

**Latent Scene Self-Refinement.** Finally, a self-attention layer Vaswani et al. (2017) is applied to the updated scene tokens to promote spatial coherence and semantic consistency:

$$\hat{\mathcal{R}}_{scene} = SelfAttn(\mathcal{R}_{scene}) \tag{13}$$

where $\hat{\mathcal{R}}_{scene}$ is the refined enhanced scene representation. This facilitates internal relational reasoning among scene elements and yields a structured representation that supports downstream high-level planning and trajectory forecasting.

### 3.6 PLANNING DECODER

The Plan Decoder generates future ego trajectories conditioned on the scene context. We initialize a set of learnable trajectory queries, $\mathbf{Q}_{wp} \in \mathbb{R}^{K \times D}$, where $K = ego\_fut\_mode \times fut\_ts$ and $D$ is the embedding dimension.

Table 1: **Comparison of state-of-the-art methods on the nuScenes dataset**. The ego status was not utilized in the planning module. ◇: Lidar-based methods. ∗: Backbone with ResNet-101 (He et al., 2016), while others use ResNet-50 or similar.∗∗: Reproducing the results of the official open-source code.†: FPS measured on an NVIDIA A100 GPU. ‡: AVG metric protocal as same as VAD.

| Method | Auxiliary Task | L2 (m) ↓ | | | | Collision Rate (%) ↓ | | | | FPS |
|---|---|---|---|---|---|---|---|---|---|---|
| | | 1s | 2s | 3s | Avg. | 1s | 2s | 3s | Avg. | |
| NMP◇ (Zeng et al., 2019) | Det & Motion | 0.53 | 1.25 | 2.67 | 1.48 | 0.04 | 0.12 | 0.87 | 0.34 | - |
| FF◇ (Hu et al., 2021) | FreeSpace | 0.55 | 1.20 | 2.54 | 1.43 | 0.06 | 0.17 | 1.07 | 0.43 | - |
| EO◇ (Khurana et al., 2022) | FreeSpace | 0.67 | 1.36 | 2.78 | 1.60 | 0.04 | 0.09 | 0.88 | 0.33 | - |
| ST-P3 (Hu et al., 2022) | Det & Map & Depth | 1.72 | 3.26 | 4.86 | 3.28 | 0.44 | 1.08 | 3.01 | 1.51 | 1.6† |
| UniAD∗ (Hu et al., 2023) | Det&Track&Map&Motion&Occ | 0.48 | 0.96 | 1.65 | 1.03 | 0.05 | 0.17 | 0.71 | 0.31 | 1.8† |
| OccNet∗ (Sima et al., 2023) | Det & Map & Occ | 1.29 | 2.13 | 2.99 | 2.14 | 0.21 | 0.59 | 1.37 | 0.72 | 2.6 † |
| VAD-Base (Jiang et al., 2023) | Det & Map & Motion | 0.54 | 1.15 | 1.98 | 1.22 | 0.04 | 0.39 | 1.17 | 0.53 | 4.5† |
| PARA-Drive (Weng et al., 2024) | Det&Track&Map&Motion&Occ | 0.40 | 0.77 | 1.31 | 0.83 | 0.07 | 0.25 | 0.60 | 0.30 | 5.0† |
| GenAD (Zheng et al., 2024b) | Det & Map & Motion | 0.36 | 0.83 | 1.55 | 0.91 | 0.06 | 0.23 | 1.00 | 0.43 | 6.7† |
| UAD∗ (Guo et al., 2024) | Det | 0.39 | 0.81 | 1.50 | 0.90 | **0.01** | 0.12 | **0.43** | **0.19** | 7.2† |
| SSR∗∗ (Li & Cui, 2025) | None | 0.25 | 0.64 | 1.33 | 0.74 | 0.07 | 0.12 | 0.74 | 0.31 | 19.6† |
| **SAR(Ours)** | Det | **0.24** | **0.63** | **1.31** | **0.73** | 0.07 | **0.11** | 0.63 | 0.27 | 8.0† |
| ST-P3‡ (Hu et al., 2022) | Det & Map & Depth | 1.33 | 2.11 | 2.90 | 2.11 | 0.23 | 0.62 | 1.27 | 0.71 | 1.6 † |
| UniAD∗‡ (Hu et al., 2023) | Det&Track&Map&Motion&Occ | 0.44 | 0.67 | 0.96 | 0.69 | 0.04 | 0.08 | 0.23 | 0.12 | 1.8† |
| VAD-Tiny‡ (Jiang et al., 2023) | Det & Map & Motion | 0.46 | 0.76 | 1.12 | 0.78 | 0.21 | 0.35 | 0.58 | 0.38 | 16.8 † |
| VAD-Base‡ (Jiang et al., 2023) | Det & Map & Motion | 0.41 | 0.70 | 1.05 | 0.72 | 0.07 | 0.17 | 0.41 | 0.22 | 4.5† |
| PARA-Drive‡ (Weng et al., 2024) | Det&Track&Map&Motion&Occ | 0.25 | 0.46 | 0.74 | 0.48 | 0.14 | 0.23 | 0.39 | 0.25 | 5.0† |
| LAW‡ (Li et al., 2024a) | None | 0.26 | 0.57 | 1.01 | 0.61 | 0.14 | 0.21 | 0.54 | 0.30 | 19.5† |
| GenAD‡ (Zheng et al., 2024b) | Det & Map & Motion | 0.28 | 0.49 | 0.78 | 0.52 | 0.08 | 0.14 | 0.34 | 0.19 | 6.7† |
| SparseDrive‡ (Sun et al., 2024) | Det & Track & Map & Motion | 0.29 | 0.58 | 0.96 | 0.61 | **0.01** | **0.05** | **0.18** | **0.08** | 9.0† |
| DriveTransformer‡(Jia et al., 2025) | Det & Map & Motion | 0.19 | 0.34 | 0.66 | 0.40 | 0.03 | 0.10 | 0.21 | 0.11 | - |
| SSR∗∗‡ (Li & Cui, 2025) | None | 0.19 | 0.36 | 0.62 | 0.39 | 0.10 | 0.10 | 0.24 | 0.15 | **19.6**† |
| **SAR‡(Ours)** | Det | **0.18** | **0.35** | **0.60** | **0.38**$^{†47\%}$ | 0.08 | 0.09 | 0.23 | 0.13$^{†41\%}$ | 8.0† |

Table 2: **Performance on the NAVSIM *navtest***. '∗' represents that the results are sourced from Hydra-MDP++ Li et al. (2025). NC: no at-fault collision. DAC: drivable area compliance. TTC: time-to-collision. Comf.: comfort. EP: ego progress. PDMS: the predictive driver model score. The best results are highlighted in bold.

| Method | Input | NC ↑ | DAC ↑ | TTC ↑ | Comf. ↑ | EP ↑ | PDMS ↑ |
|---|---|---|---|---|---|---|---|
| TransFuser Chitta et al. (2022) | C & L | 97.7 | 92.8 | 93.0 | 100 | 79.2 | 84.0 |
| LAW Li et al. (2024a) | C & L | 96.4 | 95.4 | 88.7 | 99.9 | 81.7 | 84.6 |
| GoalFlow∗ Xing et al. (2025) | C & L | 98.3 | 93.3 | 94.8 | 100 | 79.8 | 85.7 |
| DiffusionDrive Liao et al. (2024) | C & L | 98.2 | 96.2 | 94.7 | 100 | 82.2 | 88.1 |
| **SAR (Ours)** | C & L | **98.4** | **96.7** | **94.8** | 100 | **82.7** | **88.5** |

The trajectory queries are refined via cross-attention with scene-level features $R_{scene}$ :

$$\mathbf{Q}_{refined} = WayDecoder(\mathbf{Q}_{wp}, \mathcal{R}_{scene}, \mathcal{R}_{scene}). \qquad (14)$$

Finally, the refined queries are decoded into future trajectories by a MLP layer:

$$\hat{\mathbf{T}}_{ego} = EgoFutDecoder(\mathbf{Q}_{refined}) \in \mathbb{R}^{M \times T \times 2}, \qquad (15)$$

where $M$ is the number of trajectory modes, $T$ is the prediction horizon, and each trajectory consists of 2D coordinates. This design allows the Plan Decoder to leverage both global scene context and mode specific trajectory priors to generate multi-modal future paths efficiently.

## 4 EXPERIMENTS

### 4.1 BENCHMARK AND METRIC

We evaluate our SAR framework on three comprehensive benchmarks: nuScenes Caesar et al. (2020), NAVSIM Dauner et al. (2024), and Bench2Drive Jia et al. (2024).

**NuScenes Dataset and Metrics.** A standard benchmark widely adopted in autonomous driving. Rather than replicating its full-stack pipeline, we focus on two principled evaluation metrics: displacement error and collision rate. The former captures the average L2 distance between the predicted and ground-truth trajectories, indicating accuracy, while the latter quantifies safety by measuring the frequency of collisions with static or dynamic obstacles.

Table 3: **Performance on the Bench2Drive benchmark.** 'SR', 'EBrake', and 'TSign' denote the success rate, emergency braking, and traffic sign compliance, respectively. PDM-Lite is a rule-based planner that can access privileged information from the CARLA simulator.

| Method | Overall↑ | | Multi-Ability↑ | | | | | Mean ↑ |
|---|---|---|---|---|---|---|---|---|
| | DS | SR | Merge | Overtake | EBrake | GiveWay | TSign | |
| PDM-Lite Sima et al. (2024) | 97.0 | 92.3 | 88.8 | 93.3 | 98.3 | 90.0 | 93.7 | 92.8 |
| AD-MLP Zhai et al. (2023) | 18.1 | 0.0 | 0.0 | 0.0 | 0.0 | 0.0 | 4.4 | 0.9 |
| TCP Wu et al. (2022) | 40.7 | 15.0 | 16.2 | 20.0 | 20.0 | 10.0 | 7.0 | 14.6 |
| VAD Jiang et al. (2023) | 42.4 | 15.0 | 8.1 | 24.4 | 18.6 | 20.0 | 19.2 | 18.1 |
| UniAD Hu et al. (2023) | 45.8 | 16.4 | 14.1 | 17.8 | 21.7 | 10.0 | 14.2 | 15.6 |
| DriveTransformer Jia et al. (2025) | 63.5 | 35.0 | 17.6 | 35.0 | 48.4 | 40.0 | 52.1 | 38.6 |
| **SAR(Ours)** | **67.5** | **46.8** | **41.0** | **40.0** | **55.0** | **60.0** | **58.9** | **51.0** |

Table 4: **Ablation study on Scene Representation (Rep.) on NuScenes.** We compare different scene representations: *Token*: only learnable scene tokens without explicit semantic guidance; *Det*: object detection features; *Motion*: multi-agent state embedding; *Map*: HD-map lane & topology features.

| Method | Rep. | L2 (m) ↓ | | | | Collision Rate (%) ↓ | | | | FPS |
|---|---|---|---|---|---|---|---|---|---|---|
| | | 1s | 2s | 3s | Avg. | 1s | 2s | 3s | Avg. | |
| Sparse-only | Token-only | 0.26 | 0.57 | 1.01 | 0.61 | 0.14 | 0.21 | 0.54 | 0.30 | 19.5 |
| Sparse-only | Det + Map + Motion | 0.41 | 0.70 | 1.05 | 0.72 | **0.07** | 0.17 | 0.41 | 0.22 | 4.5 |
| SAR (Ours) | Token-only | 0.21 | 0.41 | 0.73 | 0.45 | 0.17 | 0.31 | 0.53 | 0.34 | - |
| SAR (Ours) | Token + Ego action | 0.20 | 0.42 | 0.67 | 0.43 | 0.09 | 0.09 | 0.30 | 0.16 | - |
| SAR (Ours) | Token + Agent action | 0.19 | 0.37 | 0.64 | 0.40 | 0.17 | 0.29 | 0.46 | 0.31 | - |
| SAR (Ours) | Token + Ego & Agent action | **0.18** | **0.35** | **0.60** | **0.38** | 0.08 | **0.09** | **0.23** | **0.13** | 8.0 |

**NAVSIM Dataset and Metrics.**    NAVSIM derived from the OpenScene Peng et al. (2023) dataset, consists of over 120 hours of realistic driving data containing high-resolution camera and LiDAR inputs over a 1.5-second time span. It filters out trivial scenes and emphasizes complex reasoning scenarios. We adopt the official PDMS for performance evaluation.

**Bench2Drive Dataset and Metrics.**    It is built upon the CARLA Dosovitskiy et al. (2017) simulator and includes 220 routes covering 44 interactive traffic scenarios under diverse weather and lighting conditions. We report both the CARLA Driving Score (DS) and multi-ability performance across tasks such as merging, overtaking, and traffic-sign compliance. Appendix shows the detailed metric definitions and implementation details.

## 4.2 COMPARISON WITH STATE-OF-THE-ART

**Results on nuScenes.**    As shown in Table 1, SAR achieves the best performance on nuScenes, outperforming all baselines in both displacement and collision metrics. It achieves an average L2 of 0.73m and CR of 0.27%, surpassing SSR (0.74m) and UAD (0.90m), while maintaining high efficiency (8.0FPS). Notably, SAR achieves strong safety without occupancy supervision and remains competitive against methods like UniAD (1.8FPS). Under the AVG protocol, SAR further leads with 0.38m average L2.

**Results on NAVSIM.**    Table 2 shows that SAR achieves the highest PDMS score (88.5), consistently outperforming strong baselines like DiffusionDrive and LAW across all sub-metrics, highlighting its robustness in long-horizon planning.

**Results on Bench2Drive.**    On the Bench2Drive dataset, SAR achieves achieves the best Driving Score (67.5) and Success Rate (46.8) in closed-loop evaluation, as shown in Table 3. It also excels in complex tasks like merging and unprotected turns. These results validate the effectiveness of our goal-driven sparse scene representation in challenging interaction scenarios.

Table 5: **Ablation on Bench2Drive dev10.**

| Method | Driving Score ↑ | Success Rate (%) ↑ |
|---|---|---|
| w/o action | 40.3 | 20 |
| w/o ego action | 52.8 | 30 |
| w/o agent action | 54.5 | 30 |
| w/ ego&agent action | **0.20** | **40** |

Table 6: **Size Configuration of SAR.**

| Method | Configuration | Driving Score ↑ | Success Rate (%) ↑ | FPS |
|---|---|---|---|---|
| SSR | 3 Layers | 45.8 | 10 | 14.2 |
| SAR-small | 3 Layers | 56.3 | 30 | 12.5 |
| SAR-base | 6 Layers | 63.1 | 40 | 7.4 |

Table 7: **Ablation on Risk.**

| Method | L2 (m) ↓ | | | | CR (%) ↓ | | | |
|---|---|---|---|---|---|---|---|---|
| | 1s | 2s | 3s | Avg. | 1s | 2s | 3s | Avg. |
| w/o risk | 0.19 | 0.36 | 0.61 | 0.39 | 0.15 | 0.16 | 0.30 | 0.20 |
| w/ dist risk | 0.18 | 0.36 | 0.62 | 0.39 | 0.11 | 0.13 | 0.25 | 0.16 |
| w/ dynamic risk | **0.18** | **0.36** | **0.60** | **0.38** | **0.08** | **0.16** | **0.23** | **0.13** |

Table 8: **Ablation on Query Formulation.**

| Query type | L2 (m) ↓ | | | | CR (%) ↓ | | | |
|---|---|---|---|---|---|---|---|---|
| | 1s | 2s | 3s | Avg. | 1s | 2s | 3s | Avg. |
| ego | 0.17 | 0.46 | 0.92 | 0.52 | 0.03 | 0.21 | 0.39 | 0.21 |
| scene | **0.18** | **0.35** | **0.60** | **0.38** | **0.08** | **0.09** | **0.23** | **0.13** |

## 4.3 Ablation Studies

**Ablation study on Scene Representation.** As shown in Table 4 SAR significantly outperforms token-only and handcrafted representations. Adding ego-action encoding notably reduces collision rate ($0.34 \rightarrow 0.16$), while agent-action features lower trajectory error ($0.45 \rightarrow 0.40$ Avg. L2). Combining both yields the best results, achieving 0.38 m Avg. L2 and 0.13% CR, a 15% trajectory error reduction and 60% fewer collisions, demonstrating the effectiveness of jointly modeling ego and agent context. An ablation analysis performed on Bench2Drive-dev10 Table 5 confirms that actions offer superior scene characterization.In Appendix A.5 we provide more ablation experiments on the hyperparameters of the components.

**Ablation study on Risk Feat.** To quantify SAR's reliance on TTC/DCPA risk features, we conducted an ablation study comparing disabled, distance-only, and full risk awareness. Results in Table 7 confirm that safety gains primarily come from time-aware TTC/DCPA assessment. Notably, SAR maintains robustness even with removed or simplified risk embeddings, mitigating concerns about catastrophic failures from faulty risk signals. Additionally, Appendix A.5 provides an analysis of the noise robustness of dynamic risk features.

**Ablation on Query Formulation.** We conducted ablation studies on query mechanisms for scene-action interaction to validate the superiority of scene-as-query. Results in Table 8 show the intuitive ego-as-query approach produces "locally optimal but globally inconsistent" decisions, compressing behavioral information into a single ego vector that struggles with scene-level consistency. In contrast, scene-as-query enables each token to function as a behavior-aware unit, building structured representations that maintain spatial-behavioral coordination and demonstrate superior consistency in complex scenarios like intersections and merging zones.

## 4.4 Study on Speed-Safety Trade-off.

The Scene-action transformer in our SAR model is built upon transformer layers. We examine how varying the number of layers affects both inference speed and safety performance, with comparative analysis between SSR and SAR architectures. As demonstrated in Table 6, while increasing model depth impacts inference latency, the substantial gains in safety metrics warrant this computational trade-off.

## 4.5 High interactive scene verification on bench2drive

To verify the SAR model's performance in high interaction scenarios, we conducted ablation experiments on 10 high interaction circuits from the bench2dirve bench. The Table 9 shows that our model significantly outperforms other sparse scene representation methods.

## 4.6 Representation Analysis of Scene Details in Action Guidance

To understand why ego and agent action modeling can enhance the representation of scene queries, we visualized the weights of scene query self-attention refinement and expanded it to a 100*100 BEV space. As shown in Fig.3, compared to the scene query attention map without action enhancement, we observe that the enhanced scene query attention map has a more detailed representation

Table 9: **High interactive scene verification on HI-B2D10.**

| Method | Overall↑ | | Multi-Ability↑ | | | | | |
|--------|----|----|-------|----------|--------|---------|-------|--------|
| | DS | SR | Merge | Overtake | EBrake | GiveWay | TSign | Mean ↑ |
| VAD Jiang et al. (2023) | 40.8 | 10.0 | 6.3 | 22.0 | 16.8 | 20.0 | 17.7 | 16.6 |
| SSR Li & Cui (2025) | 44.1 | 20.0 | 13.0 | 25.6 | 29.4 | 20.0 | 24.2 | 22.4 |
| DriveTransformer Jia et al. (2025) | 61.7 | 30.0 | 16.8 | 34.0 | 47.3 | 40.0 | 50.3 | 37.7 |
| **SAR(Ours)** | **65.7** | **40.0** | **38.9** | **37.8** | **52.9** | **60.0** | **56.7** | **49.3** |

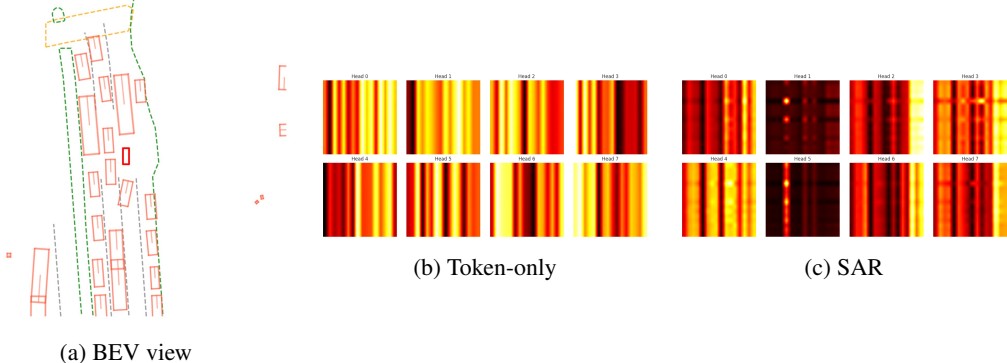

(a) BEV view     (b) Token-only     (c) SAR

Figure 3: **Visualization of Attention Maps for Different Scene Queries in the Same Scene.** Here use the weight map of 8 attention heads.

capability. It no longer focuses on the global representation, but pays more attention to key details within the region. Here we use the weight map of 8 attention heads

### 4.7 VISUALIZATION ON NUSCENES DATASET

Figure 4 shows the qualitative results of SAR trajectory planning. Experimental results demonstrate that, compared to VAD-Base, SAR maintains a safer distance from dangerous vehicles in highly interactive scenarios, ensuring greater driving safety. More results under diverse scenes are provided in Appendix A.9 due to space limitations.

## 5 CONCLUSION

We propose a framework called **SAR** for joint scene action representation. SAR leverages sparse scene tokens and ego–agent interactions to jointly encode the environment. Experimental results show that SAR outperforms existing methods across multiple standard metrics, demonstrating the effectiveness of joint scene action modeling in enabling fine grained multi agent behavior understanding and trajectory prediction. However, since our current implementation relies on single modal outputs, trajectory selection remains somewhat limited. In future work, we plan to explore multi-modal trajectory evaluation based on scene–action representations.

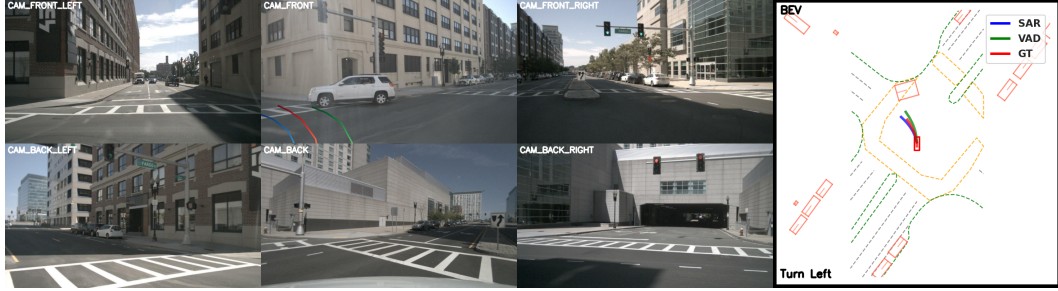

Figure 4: **Visualization of Planning Results.** The perception results are rendered from annotations.

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

# A APPENDIX

## A.1 TOKEN LEARNER IMPLEMENTATION DETAILS.

Based on the token learning mechanism described in Equation (2), this subsection elaborates on its specific implementation process. The core objective of the token learner is to adaptively select the most informative spatial locations from dense BEV feature maps, generating compact sparse scene representations. Given the input BEV feature $\mathbf{F}_{bev} \in \mathbb{R}^{H \times W \times C}$, where $H$ and $W$ represent the spatial height and width of the BEV grid respectively, and $C$ denotes the feature channel dimension, the token learner generates sparse scene tokens $\mathbf{Q}_{scene} \in \mathbb{R}^{N \times C}$ through the following steps, where $N$ is a pre-defined hyperparameter representing the number of tokens.

**Importance Score Generation:** First, a lightweight Multi-Layer Perceptron (MLP) computes scalar importance scores for each spatial position $(h, w)$:

$$\mathbf{S} = MLP(\mathbf{F}_{bev}), \quad \mathbf{S} \in \mathbb{R}^{H \times W \times 1} \tag{16}$$

The MLP consists of two fully connected layers with GELU activation functions, achieving feature channel compression and importance regression.

**Sparse Token Selection:** After flattening the spatial dimensions, Top-N selection is performed based on the importance scores:

$$\mathbf{F}_{flat} = Reshape(\mathbf{F}_{bev}) \in \mathbb{R}^{(H \cdot W) \times C} \tag{17}$$

$$\mathbf{S}_{flat} = Reshape(\mathbf{S}) \in \mathbb{R}^{(H \cdot W) \times 1} \tag{18}$$

$$\mathcal{I} = TopKIndex(\mathbf{S}_{flat}, N) \in \mathbb{R}^{N} \tag{19}$$

Here, $\mathcal{I}$ represents the index vector selecting the top $N$ positions with the highest scores from $\mathbf{S}_{flat}$.

**Scene Token Generation:** The final sparse scene tokens are formed by gathering feature vectors from the flattened features according to the selected indices $\mathcal{I}$:

$$\mathbf{Q}_{scene} = \mathbf{F}_{flat}[\mathcal{I}] \in \mathbb{R}^{N \times C} \tag{20}$$

The key advantage of this design lies in its complete differentiability, enabling end-to-end optimization of the entire token selection process. Through network training, the token learner automatically learns to allocate limited representation resources to the most informative spatial regions, such as road boundaries, traffic participants, and navigation keypoints. This adaptive selection mechanism ensures semantic richness in scene representation while maintaining computational efficiency. We also discuss the choice of N in the Appendix A.5

## A.2 LOSS & OPTIMIZATION

Our method eliminates the reliance on intermediate perception supervision and instead adopts end-to-end training using only final objective signals, including object detection and ego-trajectory regression. During training, we jointly optimize the object detection loss based on Hungarian matching (similar to DETR) and the ego-trajectory regression loss. The total loss is defined as:

$$\mathcal{L}_{total} = w_{det} \cdot \mathcal{L}_{det} + w_{traj} \cdot \mathcal{L}_{traj} \tag{21}$$

where $\mathcal{L}_{det}$ denotes the detection loss, $\mathcal{L}_{traj}$ represents the ego-trajectory regression loss, and $w_{det}$, $w_{traj}$ are task-specific weighting coefficients. This unified optimization strategy enables the model to integrate scene understanding and motion intent reasoning within a single framework, thereby enhancing its adaptability and robustness in complex traffic scenarios.

## A.3 IMPLEMENTATION DETAILS

**nuScenes.** Our SAR framework builds upon a lightweight VAD-Tiny configuration with some customized architectural modules. In open-loop settings, we adopt ResNet-50 as the image encoder and VoxelNetas the LiDAR backbone, and resize the input images to 640×360. The BEV space is discretized into a 100×100 grid and encoded as 16 sparse scene tokens of 256 dimensions. We preserve three discrete navigation commands as defined in prior works.we use 4 NVIDIA RTX H100 GPUs with a batch size of 2 per GPU and train for 12 epochs. The training time of our method is approximately 15 hours more than 10× faster than UniAD. We apply AdamW with an initial learning rate of 1e-4. Our loss function consists of three equally-weighted loss functions including a L2 trajectory loss, a cross-entropy loss for agent classification, and a smooth L1 loss for bounding box regression.

**NAVSIM.** Similar to TransFuser on the NAVSIM benchmark, we concatenate the front-view image with center-cropped front-left and front-right images, resulting in a combined resolution of 256 × 1024 pixels. For LiDAR, the point cloud in the region with 64m× 64m point cloud around the ego vehicles is used. In the network architecture, we employ ResNet34 as the backbone for BEV feature extraction similar to TransFuser. The training is conducted on the Navtrain split using 4 NVIDIA RTX H100 GPUs with a total batch size of 128, distributed across 128 epochs, and the Adam optimizer with a learning rate of 2e-4 is utilized.

**Bench2Drive** For the Bench2Drive benchmark, we use the Bench2Drive base set (1000 clips).We use ResNet50 as image backbones and the image size of (384, 1056) .The temporal length (Tqueue) are set as 3 in Bench2Drive. We use 4 NVIDIA RTX H100 GPUs with a batch size of 8 per GPU and train for 6 epochs. Training is performed using the AdamW optimizer , a weight decay of $1 \times 10^{-4}$, and a maximum learning rate of $6 \times 10^{-4}$, which follows a cosine annealing schedule for learning rate decay.

### A.4 THEORETICAL DEFINITIONS OF TTC AND DCPA

To enhance the rigor and completeness of our paper, we provide theoretical definitions for the **Time-to-Collision (TTC)** and **Distance to Closest Point of Approach (DCPA)** used in our dynamic risk-awareness model. These concepts originate from the maritime domain and are widely applied to vehicle motion planning and risk assessment.

**Time to Collision (TTC)** Time-to-Collision (TTC) measures the time required for two moving objects to collide, assuming their current relative velocities and trajectories are maintained. TTC is a predictive metric used to assess the **imminence** of a collision risk. A smaller TTC value indicates a more imminent risk, requiring faster action from the system.

Assuming the ego vehicle $p_{ego}$ and another agent $p_{agent}$ have positions $\mathbf{r}_{ego}$ and $\mathbf{r}_{agent}$, and velocity vectors $\mathbf{v}_{ego}$ and $\mathbf{v}_{agent}$ in a 2D plane, the relative position and velocity vectors are:

$$\mathbf{r}_{rel} = \mathbf{r}_{ego} - \mathbf{r}_{agent}$$
$$\mathbf{v}_{rel} = \mathbf{v}_{ego} - \mathbf{v}_{agent}$$

where $\mathbf{r}_{ego}$ and $\mathbf{r}_{agent}$ are the **position vectors** of the ego vehicle and agent, and $\mathbf{v}_{ego}$ and $\mathbf{v}_{agent}$ are their **velocity vectors**. $\mathbf{r}_{rel}$ and $\mathbf{v}_{rel}$ represent the relative position and velocity vectors.

The TTC can be calculated using the following formula:

$$\text{TTC} = -\frac{\mathbf{r}_{rel} \cdot \mathbf{v}_{rel}}{\|\mathbf{v}_{rel}\|^2} \tag{22}$$

where $\cdot$ denotes the **dot product** and $\| \cdot \|$ denotes the **Euclidean norm**.

*Note:* TTC is only physically meaningful when the relative velocity vector $\mathbf{v}_{rel}$ is directed toward the two objects approaching each other (i.e., $\mathbf{r}_{rel} \cdot \mathbf{v}_{rel} < 0$). If the objects are moving away from each other, TTC is considered infinite.

**Distance to Closest Point of Approach (DCPA)** Distance to Closest Point of Approach (DCPA) measures the minimum distance that two moving objects will reach on their future trajectories, assuming their current velocities and headings remain constant. DCPA is a predictive metric used to evaluate the **magnitude** or **severity** of a collision risk. A smaller DCPA value indicates a more dangerous potential encounter.

The calculation of DCPA is closely related to TTC. The time at which the minimum distance occurs is the TTC itself. Therefore, DCPA can be calculated using the TTC and the relative velocity vector:

$$\text{DCPA} = \|\mathbf{r}_{rel} + \mathbf{v}_{rel} \cdot \text{TTC}\| \tag{23}$$

If the calculated TTC is negative or infinite, the DCPA is simply the current distance between the two objects, $\|\mathbf{r}_{rel}\|$. In our model, DCPA and TTC work in tandem to provide a comprehensive, dynamic risk assessment.

Table 10: **Ablation on BEV Feature Modality for Agent Prediction.**

| Modality | L2 (m) ↓ | | | | CR (%) ↓ | | | |
|---|---|---|---|---|---|---|---|---|
| | 1s | 2s | 3s | Avg. | 1s | 2s | 3s | Avg. |
| Multi-modal | 0.19 | 0.40 | 0.61 | 0.40 | 0.15 | 0.29 | 0.40 | 0.28 |
| Lidar-only | **0.18** | **0.36** | **0.60** | **0.38** | **0.08** | **0.16** | **0.23** | **0.13** |

Table 11: **Ablation on Agent Supervision.**

| Percep. Sup. | L2 (m) ↓ | | | | CR (%) ↓ | | | |
|---|---|---|---|---|---|---|---|---|
| | 1s | 2s | 3s | Avg. | 1s | 2s | 3s | Avg. |
| | 0.19 | 0.36 | 0.62 | 0.39 | 0.50 | 0.58 | 0.54 | 0.54 |
| ✓ | **0.18** | **0.35** | **0.60** | **0.38** | **0.08** | **0.09** | **0.23** | **0.13** |

Table 12: **Ablation on EAD Distance**

| Dist. Param. | L2 (m) ↓ | | | | CR (%) ↓ | | | |
|---|---|---|---|---|---|---|---|---|
| | 1s | 2s | 3s | Avg. | 1s | 2s | 3s | Avg. |
| 1/3 | 0.19 | 0.39 | 0.65 | 0.41 | **0.07** | 0.13 | 0.27 | 0.16 |
| 1/4 | **0.18** | **0.35** | **0.60** | **0.38** | 0.08 | **0.09** | **0.23** | **0.13** |
| 1/5 | 0.19 | 0.40 | 0.63 | 0.41 | 0.09 | 0.14 | 0.23 | 0.15 |
| 1/6 | 0.20 | 0.43 | 0.66 | 0.43 | 0.08 | 0.13 | 0.26 | 0.16 |

Table 13: **Number of Scene Queries.**

| Number | L2 (m) ↓ | | | | CR (%) ↓ | | | |
|---|---|---|---|---|---|---|---|---|
| | 1s | 2s | 3s | Avg. | 1s | 2s | 3s | Avg. |
| 8 | 0.21 | 0.40 | 0.69 | 0.43 | 0.66 | 0.81 | 0.97 | 0.81 |
| 12 | 0.19 | 0.37 | 0.64 | 0.40 | 0.08 | 0.09 | 0.26 | 0.14 |
| 16 | **0.18** | **0.35** | **0.60** | **0.38** | **0.08** | **0.09** | **0.23** | **0.13** |
| 24 | 0.19 | 0.38 | 0.65 | 0.41 | 0.06 | 0.11 | 0.33 | 0.17 |
| 32 | 0.23 | 0.44 | 0.75 | 0.47 | 0.08 | 0.13 | 0.34 | 0.18 |

## A.5 HYPERPARAMETERS ABLATION STUDY

To evaluate the robustness of our model, we incorporate the VAD module into the LiDAR branch as the baseline version of SAR and conduct comprehensive ablation studies on the nuScenes dataset.

**Effect of BEV feature modality.** Table 10 compares the use of LiDAR-only and multimodal BEV features for agent representation. Using LiDAR-only features leads to better performance, highlighting the importance of geometric priors in behavior modeling.

**Effect of perception-level supervision.** As shown in Table 11, applying perception supervision to agent queries reduces the average L2 error from $0.39\,\mathrm{m}$ to $0.38\,\mathrm{m}$ and CR from $0.54\%$ to $0.13\%$, demonstrating its importance in learning robust and interaction-aware representations.

**Effect of center distance in EAD.** Table 12 compares different values of the center distance parameter in EAD. Setting the distance to $1/4$ (normalized by BEV grid size) yields the best performance. Smaller values restrict contextual coverage, while larger ones introduce noise, confirming that an appropriate attention range is critical for effective decoding.

**Effect of Scene Queries Number.** Table 13 discusses the number of sparse scene queries, and we find that the model achieves optimal performance when the number of scene queries is set to 16. When the number of queries is too small, it is insufficient to represent the scene semantics, while a larger number of queries leads to confusion similar to that observed when using dense feature interactions.

## A.6 STUDY ON THE NOISE ROBUSTNESS OF RISK FEATURES.

To evaluate the noise robustness of risk features, we injected Gaussian noise into the predicted trajectories required for TTC/DCPA computation during testing. Results Table 14 show that as noise intensity increases from no noise ($\sigma = 0$) to high noise levels, model performance demonstrates a gradual degradation: the average collision rate slowly rises from 0.13% to 0.17%, while the average L2 error increases from 0.38m to 0.39m. Even under the highest noise level, the system maintains stable operation without catastrophic failure, proving that while the SAR model effectively utilizes TTC/DCPA features to enhance safety, it does not exhibit over-reliance on them, demonstrating excellent fault tolerance and practical deployment reliability.

## A.7 SCENARIO-WISE ANALYSIS OF GATE WEIGHTS

We evaluated the adaptive capability of the triple-weight allocation mechanism for BEV, risk, and navigation features in our model through scenario-specific analysis and ablation studies in Fig 5. The study reveals that BEV features dominate in straightforward scenarios like car-following (0.62), navigation features gain prominence in route-planning scenarios such as intersections (0.42), and

Table 14: **Robustness to Noise in Risk Features.**

| Noise Level $\sigma$ | L2 (m) ↓ | | | | CR (%) ↓ | | | |
|---|---|---|---|---|---|---|---|---|
| | 1s | 2s | 3s | Avg. | 1s | 2s | 3s | Avg. |
| 0 (no noise) | 0.18 | 0.35 | 0.60 | 0.38 | 0.08 | 0.09 | 0.23 | 0.13 |
| low | 0.18 | 0.35 | 0.61 | 0.38 | 0.09 | 0.09 | 0.24 | 0.14 |
| medium | 0.19 | 0.34 | 0.62 | 0.38 | 0.08 | 0.10 | 0.26 | 0.15 |
| high | 0.19 | 0.35 | 0.64 | 0.39 | 0.10 | 0.12 | 0.31 | 0.17 |

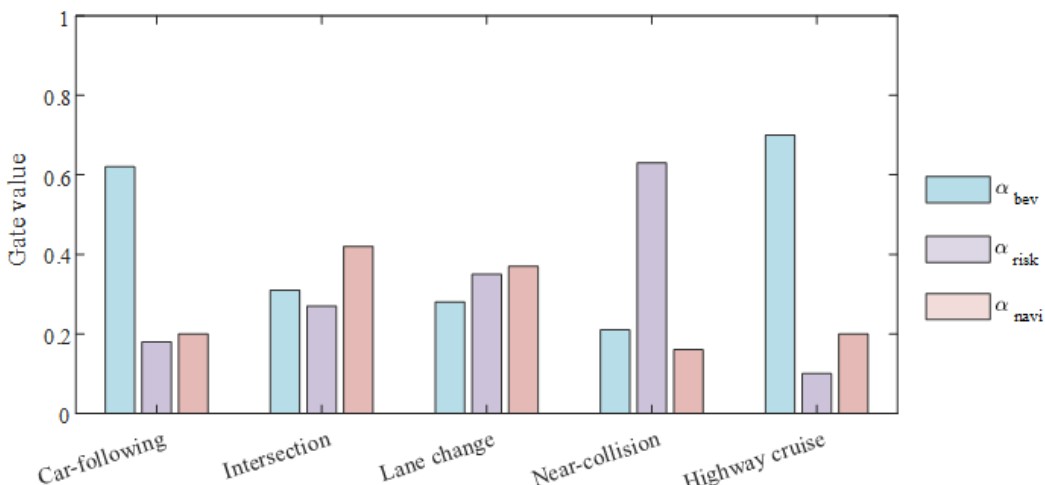

Figure 5: **Scenario-wise Analysis of Gate Weights.**

risk features surge significantly in high-risk situations like near-collision events (0.63). Ablation experiments further demonstrate that fixed-weight configurations lead to a 24% increase in average collision rate and an 18% rise in L2 error, confirming that the dynamic weight adjustment mechanism effectively adapts to diverse scenario requirements and substantially enhances system performance and safety.

## A.8 HI-B2D10 BENCHMARK

Bench2Drive consists of 220 routes. To evaluate the model's capability in representing complex traffic scenarios, we selected 10 representative high-interaction routes from Bench2Drive to construct the HI-B2D10 benchmark. Detailed route information is provided in Table 15.

## A.9 MORE VISUALIZATION

In the appendix, we provide more visualization figures in Fig. 6, Fig. 7. We also provide a demo based on the NuScenes in the supplementary materials.

## A.10 ACKNOWLEDGEMENT OF LLM USAGE

We acknowledge the use of a large language model to assist in polishing the language and improving the readability of this manuscript. No part of the scientific content, experimental design, or results was generated by the model.

Table 15: **High-Interaction Scene Verification on HI-B2D10.**

| Scenario | Route ID | Road ID | Town |
|---|---|---|---|
| MergerIntoSlowTraffic | 3048 | 639 | 12 |
| ConstructionObstacle | 24367 | 40 | 06 |
| HardBreakRoute | 24781 | 10 | 01 |
| DynamicObjectCrossing | 24211 | 6 | 01 |
| SequentialLaneChange | 17569 | 1157 | 12 |
| HighwayCutIn | 3813 | 1411 | 13 |
| NonSignalizedJunctionLeftTurn | 2084 | - | 12 |
| ParkingCrossingPedestrian | 24206 | 20 | 03 |
| InterurbanActorFlow | 24098 | 652 | 12 |
| VanillaSignalizedTurnEncounterGreenLight | 14909 | 693 | 12 |

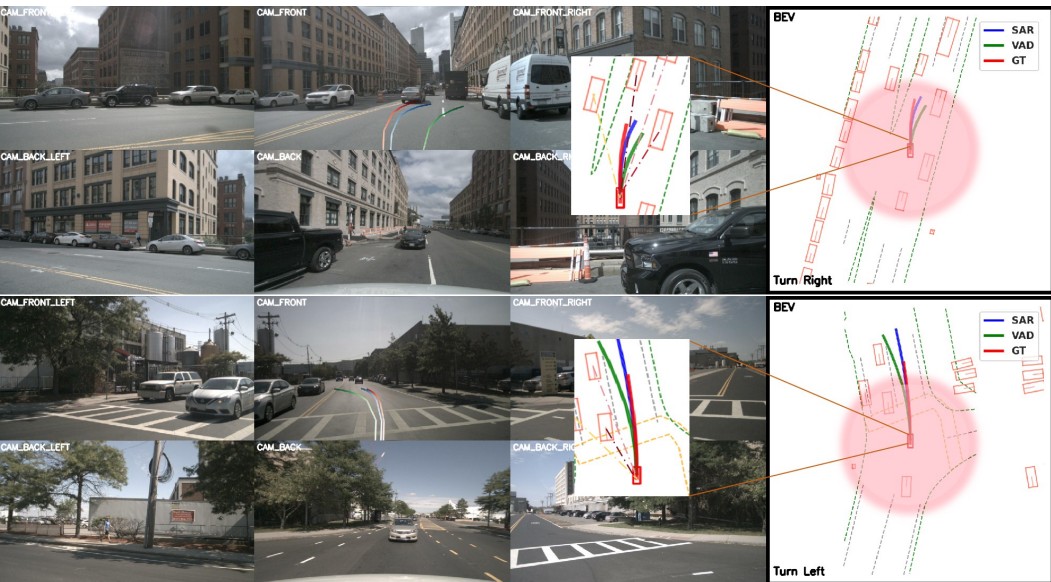

Figure 6: **Visualization of Planning Results on NuScenes.** The perception results are rendered from annotations.

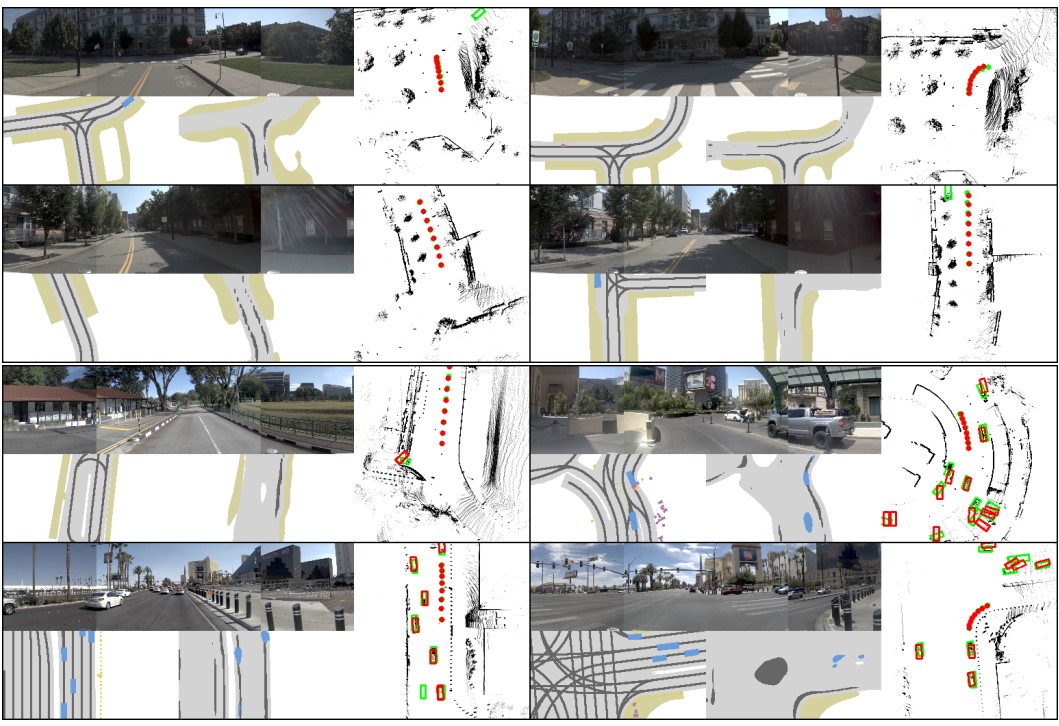

Figure 7: **Visualization of Planning Results on Navsim.** In the figure, the green trajectories represent the ground truth, while the red trajectories indicate the model predictions.

