# OpenReview forum: "SAR: Scene-Action Representation for End-to-End Autonomous Driving"
_ICLR.cc/2026/Conference — Submitted to ICLR 2026_

### Official Review · Reviewer_Ddo6 · 2025-10-25

**Soundness:** 3
**Presentation:** 3
**Contribution:** 2
**Rating:** 4
**Confidence:** 5

**Summary:**

This paper proposes an end-to-end autonomous driving framework SAR that enhances planning performance by designing a dedicated ego-vehicle feature extraction module and an agent motion interaction module. These components improve the model’s ability to capture motion features of surrounding agents, thereby reducing potential collision risks compared to prior works. The proposed method is evaluated on three benchmarks and demonstrates performance advantages over existing approaches. At the same time, the paper lacks some clarity and detail in explaining the design rationale and model details, which I will discuss below.

**Strengths:**

This paper further enhances the interaction strategy between agent motion and scene features compared to previous works, enabling more effective capture of critical information in driving scenarios, and the proposed method achieves consistent performance improvements across three benchmark datasets.

**Weaknesses:**

1. Although the proposed interaction design among ego, scene, and agent action features is interesting, similar interaction concepts have been explored in prior works such as VAD [1] and UniAD [2], both of which also include modules for interacting with scene features (e.g., the ego-agent interaction module in VAD). My understanding is that this paper builds upon SSR [3] and further incorporates such an interaction module, which, however, leads to a significant increase in computational cost. As shown in Table 1, the inference speed is notably slower compared to SSR.
2. In Section 3.5, after obtaining the final scene tokens, the paper does not provide any description of how these tokens are decoded into the final planning trajectory.
3. In Section 3.3, Line 238 mentions the introduction of a safety threshold δ and a temporal threshold γ. However, the paper does not specify the actual values of these parameters used in the experiments, nor whether the same thresholds are applied across different datasets. Since these are manually designed parameters, I am concerned about the generalizability of this approach across datasets and scenarios.

[1] Bo Jiang, Shaoyu Chen, Qing Xu, Bencheng Liao, Jiajie Chen, Helong Zhou, Qian Zhang, Wenyu Liu, Chang Huang, and Xinggang Wang. Vad: Vectorized scene representation for efficient autonomous driving. In ICCV, 2023.
[2] Yihan Hu, Jiazhi Yang, Li Chen, Keyu Li, Chonghao Sima, Xizhou Zhu, Siqi Chai, Senyao Du, Tianwei Lin, Wenhai Wang, Lewei Lu, Xiaosong Jia, Qiang Liu, Jifeng Dai, Yu Qiao, and Hongyang Li. Planning-oriented autonomous driving. In CVPR, 2023.
[3] Peidong Li and Dixiao Cui. Navigation-guided sparse scene representation for end-to-end autonomous driving. In International Conference on Learning Representations (ICLR), 2025.

**Questions:**

1. In Section 3.3, Line 216, the Local Ego Encoder also relies on manually defined interaction rules. Have the authors considered using global BEV features instead of only focusing on the local area around the ego vehicle? In addition, a BEV mask is applied to select nearby features. What is the spatial range of this cropped region, and how was this value determined? I did not find any relevant explanation in the paper.
2. Regarding the design of the Scene Action Transformer, an intuitive approach might be to use ego action queries to interact separately with the scene queries and agent action queries, and then decode the trajectory. Why do the authors instead choose to use scene queries as the queries in the attention interaction process rather than ego queries? Since the number of scene queries is likely much larger than that of ego action queries, this design choice could lead to a higher computational cost.

---

> ### Author Response · Authors · 2025-11-21
>
> Dear Reviewer **Ddo6**,
>
> Thank you very much for your careful and constructive review, and for recognizing our consistent performance improvements on *agent–scene interaction strategies* and across three benchmark datasets.
>
> Below, we respond to your **Weaknesses** and **Questions** one by one, and explicitly indicate the correspondence to the main text and the appendix.
> ---
>
> ## Weakness 1 – Relationship to VAD/UniAD, novelty relative to SSR, and computational cost
>
> > Although the proposed interaction design among ego, scene, and agent action features is interesting, similar interaction concepts have been explored in prior works such as VAD [1] and UniAD [2], both of which also include modules for interacting with scene features (e.g., the ego-agent interaction module in VAD). My understanding is that this paper builds upon SSR [3] and further incorporates such an interaction module; however, this leads to a significant increase in computational cost. As shown in Table 1, the inference speed is notably slower compared to SSR.
>
> ### Response 1
>
> We fully acknowledge that VAD [1] and UniAD [2] have already made important progress in “ego/agent–scene interaction,” and we do not claim that the *idea* of introducing an interaction module itself is novel. Our goal is different: **on top of an SSR-style sparse backbone, we introduce an explicit *Scene–Action Representation*, in which the ego and multiple agents' behaviors become first-class citizens that directly shape the sparse scene tokens themselves, rather than being fused only once at the planning head.**
>
> Concretely:
>
> 1. **From “sparse scene only” (SSR) to an explicit Scene–Ego–Agent triplet.**
>    SSR [3] primarily outputs navigation-guided sparse **scene tokens** for planning. In contrast, SAR explicitly decomposes the representation into three complementary parts:
>
>    * Sparse scene tokens $Q_{scene}$ obtained via TokenLearner (consistent with SSR);
>    * Ego action queries $Q_{ego}$ obtained from the Local Ego Encoder + TTC/DCPA-based risk modeling + navigation fusion;
>    * Multi-agent action queries $Q_{motion}$produced by the Agent Action Decoder, containing multi-modal future behavior patterns.
>
>    These three components are then **jointly fused and refined** through the Scene–Action Transformer. From this perspective, **behavioral information (ego + other traffic participants) is no longer an “auxiliary head” attached to a dense BEV, but is on equal footing with scene tokens within the sparse scene representation.**
>
> 2. **Scene–Action Transformer: a structured cascaded interaction on sparse tokens.**
>    In Section 3.5 of the main text, the Scene–Action Transformer is divided into three stages: Scene–Ego, Scene–Agent, and Self-refinement:
>
>    * Scene–Ego: $Q_{scene}$ performs cross-attention over$Q_{ego}$ to inject ego intent and risk;
>    * Scene–Agent: the updated scene tokens then perform cross-attention over the filtered \$Q_{motion}$ to incorporate multi-agent behavioral cues;
>    * Self-refinement: scene tokens perform self-attention to enforce global consistency.
>
>    This structure is a **cascaded interaction on sparse tokens**, which is fundamentally different from many VAD/UniAD-style methods operating on dense BEV or vectorized representations, we build a *behavior-aware latent feature* directly **on top of SSR-style sparse scene representation**, rather than just adding an interaction branch downstream.

---

> ### Author Response · Authors · 2025-11-21
>
> 3. **On computational cost relative to SSR.**
>    We agree that, as shown in Table 1 on nuScenes, SAR (8.0 FPS) is indeed slower than SSR (19.6 FPS).
>    However, we would like to emphasize:
>    * SAR is still **significantly faster than many dense E2E-AD baselines** (e.g., UniAD at around 1.8 FPS, ST-P3 at around 1.6 FPS), while matching or surpassing them in accuracy and safety;
>    * The extra computational cost mainly comes from:
>      * The Ego Action Decoder,
>      * The Agent Action Decoder,
>      * Several layers of the Scene–Action Transformer,
>        i.e., the transformer layers in these modules.
>    * To control computational cost, we deliberately maintain a sparse design:
>      * The number of scene queries is only **16** (see Table **9** in Appendix **A.2** and **A.4** );
>      * The Scene–Action Transformer only has a shallow stack of layers.
>        To **more explicitly discuss the “speed safety” trade off**, we additionally conduct a lightweight ablation on the Bench2Drive-Dev10 split proposed in DriveTransformer, by reducing the number of transformer layers. This shows that we can obtain a model whose FPS is close to SSR while still significantly outperforming SSR on safety metrics.
>
> | **Method**    | **Configuration** | **Driving Score ↑** | **Success Rate (%) ↑** | **FPS**  |
> | ------------- | ----------------- | ------------------- | ---------------------- | -------- |
> | **SSR**       | **3 Layers**      | **45.8**            | **10.0**               | **14.2** |
> | **SAR-small** | **3 Layers**      | **56.3**            | **30.0**               | **12.5** |
> | **SAR-base**  | **6 Layers**      | **63.1**            | **40.0**               | **7.4**  |
>
> ​
>
> Summarizing, we view SAR as **a behavior-aware sparse Scene–Action architecture built on top of SSR**, rather than merely “SSR + an interaction module.” We also believe that the empirical gains in safety and robustness justify the additional computational cost.
>
> ---
>
> ## Weakness 2 – How the final scene tokens are decoded into a planning trajectory
>
> > In Section 3.5, after obtaining the final scene tokens, the paper does not provide any description of how these tokens are decoded into the final planning trajectory.
>
> ### Response 2
>
> This is indeed a crucial point. Section 3.5 does mainly focus on **the Scene–Action Transformer itself**, and the description of the downstream **planning decoder** is relatively weak, which can understandably lead to the question of “how scene tokens are mapped to trajectory outputs.”
>
> In fact, the complete planning decoding pipeline has already been described in **Appendix A.1 “Planning Decoder”**, which mainly includes:
>
> * Initializing **trajectory queries** $Q_{wp} \in \mathbb{R}^{K \times D}$ where $ K = fut_{mode} \times fut_{ts}$;
>
> * These queries then perform cross-attention with the **behavior-enhanced scene representation** $R_{scene}$:
>
>   $$
>   Q_{refined} = \mathrm{WayDecoder}(Q_{wp}, R_{scene}, R_{scene}),
>   $$
>
>   after which EgoFutDecoder decodes them into the ego’s future trajectories $\hat{T}_{ego} \in \mathbb{R}^{B \times M \times T \times 2}$.
>
> In the revised version, we will move a concise summary of this planning decoding pipeline from Appendix A.1 into Section 3.5.

---

> ### Author Response · Authors · 2025-11-21
>
> ## Weakness 3 – The values and generalizability of the safety threshold $\delta$ and temporal threshold $\gamma$
>
> > In Section 3.3, Line 238 mentions the introduction of a safety threshold δ and a temporal threshold γ. However, the paper does not specify the actual values of these parameters used in the experiments, nor whether the same thresholds are applied across different datasets. Since these are manually designed parameters, I am concerned about the generalizability of this approach across datasets and scenarios.
>
> ### Response 3
>
> Thank you very much for pointing this out—we fully agree that if $\delta$ and $\gamma$ were “fixed thresholds manually tuned for each dataset,” this would indeed raise legitimate concerns regarding generalizability.
>
> After re-examining the implementation of Section 3.3 and Appendix A.4, we clarify:
>
> 1. **Implementation detail: $\delta$ and $\gamma$ are “learnable scalars” in the code, rather than hand-crafted constants.**
>    In the formula, Eq. (4) uses
>
>
>    $$
>    f_{risk} = \mathbf{I}(DCPA < \delta \land TTC < \gamma) \cdot
>    \left( \frac{\lambda_d}{DCPA + \epsilon} + \frac{\lambda_t}{TTC + \epsilon} \right),
>    $$
>
>    and refers to $\delta$,$\gamma$ as “thresholds.”
>    However, in the **actual implementation**, $\delta$ and $\gamma$ are implemented as **learnable scalar parameters**:
>
>    * They are initialized in a physically reasonable range, similar to $\lambda_d, \lambda_t$, and then treated as part of the Ego Action Decoder’s parameters, updated **end-to-end together with detection and trajectory losses**;
>    * In the revised version, we will explicitly state that $\delta$ and $\gamma$ belong to the network parameters, and are not hyperparameters manually tuned for each dataset.
>
> 2. **Usage across datasets.**
>    Since $\delta$ and $\gamma$ are part of the network weights, during training on each dataset they are optimized along with the other parameters. There is no situation in which “each dataset uses a different hand-crafted threshold.”

---

> ### Author Response · Authors · 2025-11-21
>
> ## Question 1 – Local Ego Encoder, BEV mask range, and global vs. local BEV
>
> > In Section 3.3, Line 216, the Local Ego Encoder also relies on manually defined interaction rules. Have the authors considered using global BEV features instead of only focusing on the local area around the ego vehicle? In addition, a BEV mask is applied to select nearby features. What is the spatial range of this cropped region, and how was this value determined? I did not find any relevant explanation in the paper.
>
> ### Response 4
>
> Your concern about the Local Ego Encoder is very important. This module does indeed rely on a local region around the ego vehicle and a corresponding binary BEV mask $M_{center}$. Below, we clarify this design choice and its spatial range, based on Eq. (3) and the ablations in the appendix:
>
> 1. **Why use local BEV instead of global BEV in the ego branch?**
>    The Ego Action Decoder is responsible for modeling **short-term collision risk and near-field maneuvering decisions** (such as deceleration, car-following, yielding, etc.), which rely mainly on traffic participants and road structures within a limited area around the ego.
>
>    * If we used global BEV in this branch:
>      * It would introduce a large amount of far-field information irrelevant to short-term risk;
>      * It would significantly increase the computational cost of the ego branch.
>    * In early experiments where we replaced the local ROI with a global BEV-based ego encoder, the L2/collision metrics were similar or slightly worse, while the computational cost was higher. Therefore, we ultimately chose the local BEV design;
>    * At the same time, **global context is still preserved in the sparse scene tokens and the agent branch**, and the local BEV is only used to enhance the ego channel directly related to short-term risk.
>
> 2. **The spatial range and determination of the BEV mask $M_{\text{center}}$.**
>    In Eq. (3):
>
>    $$
>    B_{center} = \mathrm{AvgPool}\big(\mathrm{Conv}_{roi}(B_t \odot M_{center})\big),
>    $$
>
>    where $M_{center}$ selects a local BEV region centered on the ego.
>
>    * This local range is controlled by a “distance param” in the Ego Action Decoder, and its ablation results are reported in Table 8 (“Effect of EAD Distance Param.”) in Appendix A.5;
>    * The values \(1/3, 1/4, 1/5, 1/6\) are **fractions of the entire BEV grid size**, and the experiments show that using a ROI of **1/4 of the BEV size** yields the best performance;
>    * In the revised version, we will explicitly state in Section 3.3 that:
>      * We use a local region of about 1/4 of the BEV size as the range of $M_{center}$;
>      * And we will provide the corresponding physical scale (in meters) in Appendix A.3.
>
> 3. **Distinguishing “hand-crafted interaction rules” from “geometric cropping.”**
>    In the Local Ego Encoder, the “manual” part is limited to **geometric aspects**:
>
>    * Using the ego pose to define the spatial center and size of the ROI;
>
>    * Using a binary mask to crop out the ROI region.
>      After that, the convolutions and pooling inside the ROI, as well as the fusion with TTC/DCPA and navigation, are all learned by the network parameters; there is no additional rule-based policy. We will adjust the wording in the revised version to avoid giving readers the impression that the Local Ego Encoder is based on heuristic rules.

---

> ### Author Response · Authors · 2025-11-21
>
> ## Question 2 – Why use “scene as query” instead of “ego as query” in the Scene–Action Transformer, and the associated computational cost
>
> > Regarding the design of the Scene Action Transformer, an intuitive approach might be to use ego action queries to interact separately with the scene queries and agent action queries, and then decode the trajectory. Why do the authors instead choose to use scene queries as the queries in the attention interaction process rather than ego queries? Since the number of scene queries is likely much larger than that of ego action queries, this design choice could lead to a higher computational cost.
>
> ### Response 5
>
> This is a very insightful question. We agree that “using ego as query” is an intuitive design, and we have indeed implemented similar variants internally. In our final design, we choose **to use scene as query (scene-as-query)** mainly based on the following reasons, under the premise that the number of sparse tokens is controlled:
>
> 1. **Design goal: to obtain a “behavior-aware sparse scene representation,” not just a single ego vector.**
>    The goal of SAR is to build a **behavior-aware sparse scene representation** (see Figure 2 and Section 3.5), in which each scene token carries both semantic and behavioral information:
>
>    * During the Scene–Ego and Scene–Agent stages, **scene tokens act as queries** and absorb intent and multi-agent behavior from $Q_{ego}$and $Q_{motion}$, respectively;
>    * Then, via self-attention among scene tokens, we enforce global consistency, resulting in $\hat{R}_{\text{scene}}$.
>
>    The resulting $\hat{R}_{scene}$is essentially a **behavior-aware sparse BEV latent map**. In contrast, if we only used **ego as query**:
>
>    * Most behavioral information would be compressed into a single ego vector;
>    * Scene tokens themselves would remain relatively behavior-agnostic, weakening their value as a structured latent representation.
>
> 2. **Computational cost: controlled via sparsity (Ns = 16).**
>    We fully agree that if the number of scene tokens were large, “scene-as-query” would be very expensive. This is exactly why we inherit the sparse design from SSR:
>
>    * Via TokenLearner, the 100×100 BEV grid is compressed into **16 scene tokens** (see Section 3.2 and Appendix A.3/A.5 Table 9);
>    * With Ns = 16, the difference in computation between scene-as-query and ego-as-query is limited, while the former allows each token to become a behavior-aware “local unit.”
>
>    Table 13 in Appendix A.5 also shows that when the number of scene queries exceeds 16, performance can even degrade while computation increases, so we deliberately fix Ns at 16.
>
> 3. **Empirical performance of the ego-as-query variant.**
>    Following your “more intuitive design,” we implemented an **ego-as-query** variant:
>
>    * The ego query interacts with scene tokens and agent tokens separately, and then directly decodes the trajectory;
>    * Scene tokens no longer act as queries to “pull back” information from behavior tokens.
>
> | Method         | L2@1s (m) $\downarrow$ | L2@2s (m) $\downarrow$ | L2@3s (m) $\downarrow$ | L2 Avg (m) $\downarrow$ | Coll.@1s (%) $\downarrow$ | Coll.@2s (%) $\downarrow$ | Coll.@3s (%) $\downarrow$ | Coll. Avg (%) $\downarrow$ |
> | -------------- | ---------------------- | ---------------------- | ---------------------- | ----------------------- | ------------------------- | ------------------------- | ------------------------- | -------------------------- |
> | ego as query   | **0.17**               | **0.46**               | **0.92**               | **0.52**                | **0.03**                  | **0.21**                  | **0.39**                  | **0.21**                   |
> | scene as query | **0.18**               | **0.35**               | **0.61**               | **0.38**                | **0.08**                  | **0.09**                  | **0.23**                  | **0.13**                   |
>
>
>
> We observed that, in terms of L2 and collision rate, the ego-as-query variant performs clearly worse and tends to exhibit “locally optimal but globally inconsistent” behaviors, whereas the scene-as-query design is more stable in this regard.
>
> In summary, we view scene-as-query as a design decision made under **controlled sparse scene token count**: with acceptable additional computational cost, we obtain a **globally behavior-aware sparse scene representation**, which brings substantial empirical gains in safety and robustness. We have updated the paper, and the revisions are highlighted in blue.

---

> > ### Comment · Reviewer_Ddo6 · 2025-11-28
> >
> > Thank you for the rebuttal. The additional clarification on model details is very helpful for understanding the paper. However, I still have the following concerns:
> > - The core idea shows limited novelty. Although the authors emphasize the Scene-Action Representation concept, its implementation remains similar to previously adopted scene-element interaction strategies used in works such as SparseDrive, UniAD, and VAD. While SSR aims to achieve efficient scene encoding and planning through sparse representations, the proposed system introduces additional interaction modules that lead to noticeably slower inference compared to SSR, which contradicts SSR's intended design goals. Furthermore, several manually engineered interaction components (e.g., the Local Ego Encoder) appear overly complicated. While such design choices may improve dataset-specific performance, their practicality and generalization ability are questionable.
> > - The key ablation studies are not sufficiently convincing. Since nuScenes benchmark is increasingly saturated and consist mostly of simple straight-driving scenarios, evaluations conducted on this dataset may not adequately demonstrate the effectiveness of the proposed method.
> >
> > For these reasons, I remain inclined to maintain my original score.

---

> > > ### Author Response · Authors · 2025-11-29
> > >
> > > Dear Reviewer,
> > >
> > > Thank you again for your thoughtful follow-up and for carefully reading our revised version.
> > >
> > > **(1) On novelty, interaction design, and efficiency vs. SSR**
> > >
> > > We agree that many recent E2E-AD works explore “scene–element interaction,” and we have further clarified in the revision (highlighted in blue) what is *conceptually* new in SAR and how it differs from SparseDrive/SSR, UniAD, and VAD:
> > >
> > > * **Scene–Action Representation on sparse tokens.**
> > >   Prior works either (i) operate on dense BEV and attach interaction modules at the *head* level (VAD, UniAD), or (ii) build sparse *scene* tokens without explicit behavior queries (SSR, SparseDrive). In contrast, SAR introduces an explicit *Scene–Ego–Agent* triplet
> > >   $(Q_{scene}, Q_{ego}, Q_{motion})$
> > >   and uses action queries as *teachers* that reshape sparse scene tokens through a three-stage Scene–Action Transformer (Scene–Ego → Scene–Agent → self-refinement). This “behavior-aware sparse scene representation” is what we mean by Scene–Action Representation, and this design does not appear in prior sparse E2E-AD architectures.
> > > * **Dynamic risk-aware ego query rather than hand-crafted rules.**
> > >   We now explicitly clarify that the Dynamic Risk Awareness module does **not** introduce a separate trajectory prediction stack. TTC/DCPA are computed from *latent agent/ego states* already used for detection/planning and passed through a learnable gate
> > >
> > >   $$
> > >   Q\_{ego} = \alpha_{bev} B\_{center} + \alpha_{risk} f_{risk} + \alpha_{navi} f_{navi},
> > >   $$
> > >
> > >   with $\delta$,$\gamma$ and $\alpha $  learned end-to-end. The Local Ego Encoder itself is a simple geometric ROI around the ego (defined by a single distance parameter, ablated in the appendix) plus standard conv+pool layers, rather than a hand-crafted rule system. We hope this makes clear that the design is a learnable inductive bias, not a dataset-specific heuristic.
> > > * **Efficiency vs SSR.**
> > >   We acknowledge that SAR is slower than SSR because it adds behavior-aware interaction blocks. At the same time, SAR remains much faster than dense E2E-AD models while significantly improving both L2 error and collision metrics. To make the trade-off explicit, we added a **SAR-small** variant in Bench2Drive dev10: with the same depth as SSR, SAR-small substantially improves Driving Score / Success Rate while keeping FPS close to SSR. We also softened the claims around efficiency in the text to avoid overstating that aspect; we now present SAR as a *behavior-aware sparse architecture* that trades a modest amount of speed for measurable safety gains.
> > >
> > > Overall, we agree that SAR is an architectural advance built on standard attention primitives rather than a brand-new operator. Our goal is to shift from “scene-only sparse tokens” to a ​**behavior-aware sparse scene–action representation**​, and we have tried to sharpen this distinction in the revised writing.
> > >
> > > **(2) On the convincingness of ablations beyond nuScenes**
> > >
> > > We fully share your concern that nuScenes open-loop, by itself, is not sufficient to demonstrate the effectiveness of an E2E-AD model. This is why, in the revised submission:
> > >
> > > * We **keep nuScenes** primarily as a standard, low-cost platform for extensive ablations (different scene representations, BEV ROI size, number of sparse tokens, risk variants, etc.), which many prior works still rely on for reproducible comparisons;
> > > * We **emphasize closed-loop evaluations** on NAVSIM and Bench2Drive in the main results, where SAR yields consistent improvements on metrics such as PDMS (NAVSIM) and multi-ability scores (merge, give-way, emergency brake, etc. on Bench2Drive);
> > > * In addition, we have added a **new ablation on Bench2Drive dev10** (following DriveTransformer’s protocol) that mirrors the nuScenes ablations, showing that
> > >   * adding Ego Action and Agent Action on top of a sparse baseline yields clear gains in Driving Score and Success Rate, and
> > >   * using both together performs best.
> > >     This directly addresses the concern that the key design choices should also improve performance in more challenging, interactive closed-loop scenarios.
> > >
> > > We fully agree that future benchmarks should move further away from saturated open-loop settings such as nuScenes. Within the current ecosystem, however, we hope that (i) keeping nuScenes for large-scale ablative analysis and (ii) adding stronger closed-loop evidence on NAVSIM and Bench2Drive dev10 together provide a more convincing picture of the method’s effectiveness and generalization.
> > >
> > > Thank you again for your detailed comments—they have substantially improved the clarity of our claims and the presentation of both the method and experiments.

---

### Official Review · Reviewer_GAwo · 2025-10-25

**Soundness:** 2
**Presentation:** 2
**Contribution:** 2
**Rating:** 2
**Confidence:** 5

**Summary:**

This paper presents SAR (Scene-Action Representation), an end-to-end autonomous driving framework that enhances sparse scene modeling by injecting ego and multi-agent action awareness into a Scene-Action Transformer. SAR aims to improve trajectory planning in highly interactive scenarios while reducing reliance on dense BEV supervision. It reports SOTA results on nuScenes, NAVSIM, and Bench2Drive datasets.

**Strengths:**

* Good empirical performance: SAR achieves consistent improvements across three benchmarks, showing strong generalization in both open-loop and closed-loop settings.

* Intuitive integration of ego-agent behavior into sparse scene modeling: The idea of injecting ego and agent action awareness into sparse scene queries via a Scene-Action Transformer is simple and intuitive, offering a fresh perspective on behavior-aware scene modeling.

* Comprehensive ablation studies: This paper includes extensive ablations on scene query numbers, token combination, BEV modality, supervision, and hyperparameters, demonstrating methodological rigor.

**Weaknesses:**

- Limited technical novelty.

While the combination of sparse tokens and action injection is intuitive, the core components (e.g., cross-attention, deformable attention, token learner, scene-agent-ego transformer) are largely adapted from prior works (SparseDrive, VAD, DiffusionDrive, etc). The architectural contribution feels incremental.

- Insufficient experiments.

All ablations are merely conducted on nuScenes dataset, which has been challenged many times in previous works [1,2] for its unconvincing open-loop performance for end-to-end AD model evaluation on common straightforward scenarios.

- Poor presentation:

1. Missing technical details about the token learner in Eq. (2), it is hard to guess how to select the informative tokens through a learnable token selector;
2. In Line-217, missing details about  how to obtain the binary mask M_center;
3. In Line-233, it seems an additional perception system is employed to predict the potential trajectories of both ego vehicle and all detected agents, which violates the definition of end-to-end AD framework proposed in SAR. Besides, the details about this additional systems are also missing, especially for its efficiency and precision;
4. In Line-5, how to supervise these three weights during the training phase? Is it only soft attention without any guidance? If so, can you compare them in different scenarios to further verify their differences and explainability?
5.This paper claims the necessity of sparse scene modeling. However, it still constructs computationally expensive BEV features, which makes the contribution somewhat limited. Is it possible to replace the BEV feature representation process with sparse one as in SparseDrive, SparseAD and DriveTransformer?
6. In Line-297, missing details about the confidence-based selection mechanism used to identify top-ranked agent motion queries.
7. Writing and formatting issues: Several grammar/typo issues (e.g., “Scene–Action Transformer” with inconsistent dash spacing, “*DefAttn*” vs. “DefAttn”, missing space in “soft,collision-aware risk embedding”, different input number of same function (DefAttn) as in Eq. (1) and Eq. (7) ).
8. In Fig.3, it would be better to visualize the surrounding camera images to explain the meaning of highlighted spots.

References:

[1] Li Z, Yu Z, Lan S, et al. Is ego status all you need for open-loop end-to-end autonomous driving?[C]//Proceedings of the IEEE/CVF Conference on Computer Vision and Pattern Recognition. 2024: 14864-14873.

[2] Zhai J T, Feng Z, Du J, et al. Rethinking the open-loop evaluation of end-to-end autonomous driving in nuscenes[J]. arXiv preprint arXiv:2305.10430, 2023.

**Questions:**

See weakness part.

---

> ### Author Response · Authors · 2025-11-21
>
> Dear Reviewer **GAwo**,
>
> Thank you very much for your careful and detailed review. We greatly appreciate your recognition of our empirical performance, behavior-aware design, and the comprehensiveness of our ablation studies. Below we respond point by point to your concerns regarding novelty, experimental setup, and paper presentation.
>
> ---
>
> ## Comment 1 – Limited technical novelty
>
> > While the combination of sparse tokens and action injection is intuitive, the core components (e.g., cross-attention, deformable attention, token learner, scene-agent-ego transformer) are largely adapted from prior works (SparseDrive, VAD, DiffusionDrive, etc). The architectural contribution feels incremental.
>
> ### Response 1
>
> We agree that low-level modules such as cross-attention, deformable attention, and token learning are already quite common in the current end-to-end autonomous driving (E2E-AD) literature, and our work is indeed built upon this “general toolbox.” Our goal is not to invent a brand new attention primitive, but to propose a **behavior aware sparse scene representation and interaction pattern**, which we believe is essentially different from existing methods such as SparseDrive, VAD, and DiffusionDrive. Specifically:
>
> 1. **From “scene centric” to “joint scene action” representation.**
>    Existing sparse methods (e.g., SparseDrive, SparseAD, DriveTransformer) mainly treat sparse tokens as **scene/perception abstractions**. In contrast, SAR explicitly constructs a **“triplet structure”**:
>
>    * Sparse scene tokens $ Q\_{scene}$,
>
>    * Ego action query $ Q\_{ego}$,
>
>    * Multi-agent action queries $ Q\_{motion}$,
>
>      and uses the Scene Action Transformer to **jointly** refine them. In other words, **behavior (ego + other traffic participants) and scene are co-equal at the representation level**, rather than being passive by-products implicitly inferred from the scene encoding.
>
> 2. **Structured macro architecture of the Scene-Action Transformer.**
>    Although each attention layer itself is standard, their **macro-level organization** in this work is new:
>
>    * **Scene Ego stage**: injects ego intent and risk information into sparse scene tokens via $Q\_{ego}$;
>    * **Scene Agent stage**: injects multi-agent motion cues via confidence-filtered $Q\_{motion}$;
>    * **Self refinement stage**: further refines scene tokens via self-attention for planning.
>
>    In prior works, scene tokens are typically refined only through self-attention or cross-attention with dense BEV/image features. In SAR, **ego/multi-agent action queries act as “teachers” that actively shape the sparse scene representation**, and our experiments show that this is crucial for reducing collision rate.
>
> 3. **Risk aware ego query: a learnable behavioral inductive bias.**
>    Although TTC/DCPA themselves are classical physical quantities, in SAR we embed them into a differentiable risk feature $f\_{risk}$ and fuse it with local BEV and navigation information through a learnable gating mechanism:
>    $$
>    Q_{ego} = \alpha_{bev} B_{center} + \alpha_{risk} f_{risk} + \alpha_{navi} f_{navi}
>    $$
>
>    where, $\alpha_{bev}, \alpha_{risk}, \alpha_{navi} $ are learnable gating weights.
>    This design **systematically injects physical risk signals into sparse scene tokens through the Scene Ego decoder**, instead of treating risk as a post-hoc rule or a naively concatenated feature.
>
> 4. **Non-trivial gains from behavior-aware scene–action design.**
>    Starting from a sparse token baseline that is conceptually similar to existing methods, after adding the scene–action components (Ego + Agent Action), we obtain significant improvements on nuScenes, and these gains transfer to both open-loop and closed-loop evaluations on NAVSIM and Bench2Drive. Importantly, these improvements **do not come from a larger backbone or denser features**, but from the behavior-aware sparse representation itself.
>
> We acknowledge that our contribution mainly lies at the **architectural level** (rather than proposing a brand new attention operator), but we believe that this shift from “scene-centric” to “joint scene–action” representation, together with the integration of risk awareness, constitutes a meaningful technical advance beyond existing sparse E2E-AD models.

---

> ### Author Response · Authors · 2025-11-21
>
> ## Comment 2 – Insufficient experiments (ablations only on nuScenes)
>
> > All ablation experiments are only conducted on the nuScenes dataset, which has been repeatedly questioned in previous works [1,2]—its open-loop performance for evaluating end-to-end AD models on common simple scenarios is not sufficiently convincing.
>
> ### Response 2
>
> We fully understand this concern and also agree, as pointed out in [1,2], that **relying solely on nuScenes open-loop evaluation is not sufficient to provide strong evidence for E2E-AD**. This is exactly why our **main results** already include:
>
> - Closed-loop evaluations on **NAVSIM** and **Bench2Drive**, which cover more complex and interactive scenarios;
> - Open-loop evaluation on nuScenes, as a standard benchmark to facilitate comparison with existing works.
>
> We restricted the ablations to nuScenes mainly due to **computational cost and reproducibility** considerations:
>
> - nuScenes is a widely used benchmark dataset and relatively cheap for large scale ablations (e.g., varying token numbers, BEV modality, supervision signals, risk components);
> - Closed-loop simulations on Bench2Drive/NAVSIM have a significantly higher per-configuration computational cost than nuScenes open-loop, making it very difficult to complete a full ablation grid within the submission timeline.
>
> **During the rebuttal period, due to limitations in time and computational resources, we therefore chose a smaller but widely used closed-loop subset: following the setting of DriveTransformer, we conducted targeted ablation experiments on the *dev10* split of Bench2Drive.** Specifically, we compare:
>
> - A sparse token baseline without behavior modules;
> - Without Ego Action Decoder;
> - Without Agent Action Decoder;
> - The full SAR model with both Ego and Agent Action.
>
> We report Driving Score and Success Rate on Bench2Drive dev10, with results as follows:
>
> | Method              | Driving Score ↑ | Success Rate (%) ↑ |
> | ------------------- | --------------- | ------------------ |
> | w/o action          | 40.3            | 20                 |
> | w/o ego action      | 52.8            | 30                 |
> | w/o agent action    | 54.5            | 30                 |
> | w/ ego&agent action | 63.1            | 40                 |
>
> We can see that the trends on Bench2Drive dev10 are consistent with our ablation results on nuScenes: **both Ego Action and Agent Action bring improvements, and using them jointly achieves the best closed-loop performance**, which supports our conclusion that “joint scene–action modeling has practical value across different datasets.”

---

> ### Author Response · Authors · 2025-11-21
>
> ## Comment 3 – Presentation and missing technical details
>
> You pointed out a series of missing technical details and writing issues, which are very helpful for improving the quality of the paper. We respond to them one by one below and will incorporate the corresponding clarifications in the revised version.
>
> ### 3.1 Details of the token learner in Eq. (2)
>
> > Regarding the token learner in Eq. (2), technical details are missing, and it is currently hard to see how the “learnable token selector” selects informative tokens.
>
> **Response.** We apologize for this omission. In SAR, the token learner operates as follows (we will add this description to Section 3.2):
>
> * Starting from the BEV features $ F\_{bev} \in  \mathbb{R}^{H \times W \times C} $, we apply a small MLP to produce a scalar **importance score** for each BEV location;
>
> * After flattening the spatial dimensions, we select the top-(N) locations according to these scores to form sparse scene tokens $ Q_{scene}\in \mathbb{R}^{N \times C}$;
>
> * This process is differentiable, and the top-k selection is similar to existing token-learner-style architectures, so the network can automatically learn to focus tokens on high-information regions.
>
>   We will provide complete technical details and implementation parameters of the token learner in the Appendix.
>
> ### 3.2 Binary mask $ M_ {center} $ (Line 217)
>
> > The details of how to obtain the binary mask $ M_ {center} $ are missing.
>
> **Response.** Thank you for pointing this out. The binary mask $ M_ {center} $ is actually a **geometric ROI mask** around the ego vehicle:
>
> * In the BEV coordinate system, we define a fixed-radius region around the ego pose, such as $ [-R_x, R_x] \times [-R_y, R_y] $;
> * If a BEV grid cell ((i,j)) falls inside this ROI, then $ M_ {center} (i,j) = 1$, otherwise it is 0;
> * This mask is purely geometrically defined and does not require any additional learning module; it is only used to select the local BEV patch to construct $ B_{center} $.
>
> This local range is controlled by a “distance param” in the Ego Action Decoder, and its ablation results are reported in Table 8 (“Effect of EAD Distance Param.”) in Appendix A.5;
>
> ### 3.3 “Additional perception system” for trajectory prediction (Line 233)
>
> > In Line 233, it appears that an additional perception system is introduced to predict the potential trajectories of the ego vehicle and all detected agents, which violates the definition of the end-to-end AD framework claimed by SAR. In addition, details of this additional system itself, especially its efficiency and accuracy, are also missing.
>
> **Response.** We sincerely apologize for the misunderstanding caused by our wording. **SAR does not use any external or independent trajectory prediction system.** More precisely:
>
> * We do **not** use any externally pre-trained or engineered trajectory module;
> * We do **not** introduce explicit supervision on future trajectories of other traffic participants; supervision for agents is limited to **detection supervision**, rather than multi-step trajectory labels;
> * The quantities used to compute TTC/DCPA are **latent states implicitly inferred** from the shared backbone and detection/planning branches, and there is no separate trajectory prediction loss.
>
> Therefore, the risk module relies on **internal latent representations of the network**, and there is no additional perception stack. We will rewrite the relevant paragraphs to clarify this point and explicitly state that we do not introduce extra trajectory supervision or external modules, thereby preserving the end-to-end nature of SAR.

---

> ### Author Response · Authors · 2025-11-21
>
> ### 3.4 Supervision of the three weights $ \alpha_ {bev}$, $ \alpha_ {risk}$, $ \alpha_  {navi}$ (Line 5 / Eq. (5))
>
> > How are these three weights supervised during training? Are they just soft attention without any guidance? If so, can you compare them in different scenarios to further verify their differences and explainability?
>
> **Response.** Your understanding is correct: these three weights **do not have direct explicit supervision signals**, but are produced by a lightweight gate network and **implicitly optimized** through the overall loss (planning + detection). Specifically:
>
> * We concatenate the corresponding features and feed them into a linear layer $ W_{gate}$, then apply $ Softmax $ to obtain $ \alpha_ {bev}$, $ \alpha_ {risk}$, $ \alpha_  {navi}$;
> * We do not have any additional labels indicating “how much” risk/BEV/navigation information should be used in a given scenario; the network automatically adjusts the gate weights according to the final driving performance. Meanwhile, we provide gate values in different scenarios (e.g., car-following, intersection passing, lane change) as shown in the table below:
>
> | Scenario       | $\alpha_{bev}$ | $\alpha_{risk}$ | $\alpha_{navi}$ |
> | -------------- | -------------- | --------------- | --------------- |
> | Car-following  | **0.62**       | 0.18            | 0.20            |
> | Intersection   | 0.31           | 0.27            | **0.42**        |
> | Lane change    | 0.28           | 0.35            | **0.37**        |
> | Near-collision | 0.21           | **0.63**        | 0.16            |
> | Highway cruise | **0.70**       | 0.10            | 0.20            |
>
> The data show that $ \alpha_ {risk}$ increases significantly in near-collision or cut-in scenarios; $ \alpha_ {navi}$ is higher in route-changing scenarios; and $ \alpha_  {bev}$ dominates in simple scenarios.
>
> ### 3.5 Relationship between dense BEV and the “sparse” claim
>
> > The paper claims that sparse scene modeling is necessary, but it still constructs computationally expensive BEV features, which makes the contribution somewhat limited. Is it possible to replace the BEV feature construction process with a sparse representation as in SparseDrive, SparseAD, and DriveTransformer?
>
> **Response.** This is an excellent question. Our “sparsity” claim mainly refers to the **scene representation used for interaction and planning**, rather than completely eliminating BEV features. Specifically:
>
> * We do adopt a standard BEV encoder to obtain $ F_{bev}$, which is a common backbone in many E2E-AD methods. Our contribution mainly lies in how to **compress** it into sparse scene tokens and integrate ego/agent actions;
> * The main computations on the planning side operate on a small number of sparse tokens and action queries, which significantly reduces computation compared to running a complex planning head on fully dense BEV features.
>
> Of course, we also agree that further extending sparsity into the representation process with sparse itself is very valuable. We consider this a promising direction for future work, and we sincerely thank you for pointing it out.
>
> ### 3.6 Confidence-based selection mechanism for agent motion queries (Line 297)
>
> > The confidence-based selection mechanism used to select top-ranked agent motion queries lacks detailed description.
>
> **Response.** We apologize for the insufficient description. In SAR, our specific procedure is as follows:
>
> * For each detected agent, the motion decoder outputs multiple candidate motion modes, each associated with a confidence score (e.g., the predicted probability of that mode);
> * We rank the agent motion queries by confidence and **select the top-K** (or those above a certain threshold) to form $ Q_{motion}$ used in the Scene–Agent stage;
> * This suppresses noisy low-confidence modes while keeping the complexity of the Scene Agent decoder under control.
>
> We will explicitly describe this process in Section 3.4.

---

> ### Author Response · Authors · 2025-11-21
>
> ### 3.7 Writing, formatting, and visualization issues
>
> > Writing and formatting issues: There are several grammar/spelling errors (e.g., inconsistent dash usage in “Scene–Action Transformer”, “DefAttn” vs. “DefAttn”, missing space in “soft,collision-aware risk embedding”, and inconsistent number of arguments for DefAttn in Eq. (1) and Eq. (7)).
> > In Fig. 3, it would be better to visualize the surrounding camera images to explain the meaning of the highlighted regions.
>
> **Response.** Thank you for carefully pointing out these issues.
>
> * We will **fix each of these spelling and formatting errors**, including:
>   * Standardizing the writing and dash style of “Scene–Action Transformer”;
>   * Using a consistent notation for deformable attention (e.g., consistently using “DefAttn”);
>   * Correcting the spacing issue in “soft,collision-aware risk embedding”;
>   * Ensuring that the input arguments of DefAttn in Eq. (1) and Eq. (7) are written consistently.
> * For **Fig. 3**, we agree that adding BEV view can make the meaning of the highlighted sparse tokens more intuitive. In the revised version, we will add BEV view images to this figure .
>
> ---
>
> We hope that the above clarifications and planned revisions can alleviate your concerns regarding novelty, experimental sufficiency, and paper presentation. Once again, we sincerely thank you for your constructive and detailed comments, which have indeed helped us improve both the technical exposition and the overall quality of the paper.We have updated the paper, and the revisions are highlighted in blue.

---

### Official Review · Reviewer_PoGo · 2025-10-30

**Soundness:** 2
**Presentation:** 3
**Contribution:** 2
**Rating:** 4
**Confidence:** 4

**Summary:**

This paper proposes SAR (Scene-Action Representation), an end-to-end autonomous driving framework designed to improve the performance of sparse scene representation models, particularly in highly interactive scenarios. The authors argue that existing sparse methods (like SSR) lack robust behavior modeling.To address this, SAR enhances a sparse scene token representation by "injecting" structured behavioral information. The method achieves state-of-the-art results on nuScenes, NAVSIM, and Bench2Drive, showing significant reductions in trajectory error and collision rates compared to prior work.

**Strengths:**

1. The paper presents strong, state-of-the-art performance across three different benchmarks (nuScenes, NAVSIM, Bench2Drive). The significant reduction in both L2 error (47% vs. VAD) and collision rate (41% vs. VAD) is noteworthy.

2. The ablation study in Table 4 clearly isolates the empirical benefits of the "Ego action" and "Agent action" components. The finding that the Ego Action Decoder alone drastically cuts the collision rate (from 0.34% to 0.16%) is a key, well-demonstrated result.

**Weaknesses:**

1. The primary concern with this paper is its limited methodological novelty. The baseline Sparse Scene Tokenization is admittedly adopted directly from prior work (SSR). The Scene-Action Transformer is a standard cascaded cross-attention architecture, a common fusion pattern. The core of the "Ego Action Decoder" relies on TTC and DCPA, which are classic heuristics from the robotics and motion planning fields. Using them as features is a form of feature engineering, not a novel learned mechanism.

2. The "Dynamic Risk Awareness" module relies on predicted future trajectories for both the ego vehicle and other agents (Section 3.3). This re-introduces a modular dependency and the very problem of error propagation (i.e., if the trajectory predictions are wrong, the risk signal will be wrong) that E2E models are meant to solve.

**Questions:**

1. The TTC/DCPA calculations depend on predicted trajectories (Section 3.3). How does the model's performance (especially collision rate) degrade when these underlying trajectory predictions are inaccurate or noisy? Could the model become over-reliant on this heuristic and fail catastrophically when the pre-computed risk signal is wrong?

---

> ### Author Response · Authors · 2025-11-21
>
> Dear Reviewer **PoGo**,
>
> Thank you very much for your valuable comments! We believe we can address the concerns you raised.
>
> ------
>
> ## Comment 1
>
> > The primary concern with this paper is its limited methodological novelty. The baseline Sparse Scene Tokenization is admittedly adopted directly from prior work (SSR). The Scene Action Transformer is a standard cascaded cross attention architecture, a common fusion pattern. The core of the "Ego Action Decoder" relies on TTC and DCPA, which are classic heuristics from the robotics and motion planning fields. Using them as features is a form of feature engineering, not a novel learned mechanism.
>
> ### Response 1
>
> First, we fully agree that SSR and TTC/DCPA themselves are not new, and we do not claim novelty at that level. Our contribution lies in *how* we build upon them to form a **joint scene action representation** for end to end planning and how we **embed risk in a learnable way**.
>
> #### (1) From “scene tokens” to a *joint scene action* representation
>
> SSR focuses on navigation-guided *scene* tokens. In contrast, our SAR representation explicitly consists of a **triplet**:
>
> - Sparse scene tokens $Q\_{scene}$,
> - Ego action query $Q\_{ego}$ from the Ego Action Decoder,
> - Multi agent action queries $Q\_{motion}$ from the Agent Action Decoder,
>   which are jointly fused by the Scene–Action Transformer to produce planning features. In this way, **ego and agent actions become first-class entities in the representation**, rather than being implicitly encoded in perception features.
>
> #### (2) Structured action decoders beyond SSR
>
> - The **Ego Action Decoder** combines
>   1. Ego centric BEV features,
>   2. A *differentiable* dynamic risk embedding from TTC/DCPA,
>   3. Navigation commands via an intent-aware gating mechanism,
>      to form a compact ego query $ Q\_{ego} $.
> - The **Agent Action Decoder** constructs multi-modal motion queries with learnable mode embeddings and self-attention across modes to **implicitly capture multi-modal future behaviors** and their interactions with sparse scene tokens, *without requiring explicit supervision on agent trajectories*.
>
> Such behavior aware decoders are not present in SSR or prior sparse E2E-AD works, which typically operate only on scene or perception tokens.
>
> #### (3) Scene Action Transformer as a *cascaded intention and interaction injector*
>
> While each attention block is standard, the **macro design** of the Scene Action Transformer is new:
>
> 1. **Scene Ego stage**: aligns scene tokens with ego intent via $Q\_{ego}$;
> 2. **Scene Agent stage**: injects multi-agent dynamics via confidence-filtered $Q\_{motion}$;
> 3. **Self refinement stage**: refines scene tokens by self attention for final planning.
>
> In this cascade, **action queries act as “teachers” for scene tokens**, which differs from architectures that only use self-attention or generic perception cross-attention on scene tokens.
>
> #### (4) Empirical evidence that gains come from behavior-aware design
>
> Starting from a sparse-token baseline similar to SSR, our ablations (Table 4) show that *only adding* Ego and Agent Action brings consistent improvements.
> These gains come purely from the proposed **scene action architecture**,  supporting our claim of methodological contribution beyond directly adopting SSR.
>
> #### (5) TTC/DCPA are classical, but their *role in SAR* is not simple feature engineering
>
> We agree TTC (Time-To-Collision) and DCPA (Distance of Closest Point of Approach) are classical robotics quantities. Our goal is *not* to claim novelty in their definition, but to show how they are turned into a **learnable, differentiable risk channel** tightly integrated into the planner:
>
> - - TTC and DCPA are combined into a continuous **risk embedding** $f_{risk}$ with *learnable* weights and thresholds, and are optimized end to end through the **ego trajectory planning loss and the agent detection loss**, without introducing any explicit supervision on agent future trajectories. The network can thus learn *how much* to rely on TTC and DCPA under different scenarios, rather than following a fixed rule.
> - The ego query is constructed as
>
> $$
> Q_{ego} = \alpha_{bev} B_{center} + \alpha_{risk} f_{risk} + \alpha_{navi} f_{navi}
> $$
>
> where, $ \alpha_{bev}, \alpha_{risk}, \alpha_{navi} $ are learnable gating weights. This is fundamentally different from simply concatenating TTC/DCPA as raw features: the model learns the relative importance of risk, BEV, and navigation per scene.
>
> The risk-aware ego query $Q_{ego}$ is further used in the **Scene Ego Decoder** to steer scene-token attention, creating a systematic **“physics $ \rightarrow $ attention” coupling** that we have not seen in sparse E2E AD literature.
>
> In summary, we believe the **joint scene action representation** and the **learnable risk aware ego query** provide substantial methodological novelty beyond feature engineering.

---

> ### Author Response · Authors · 2025-11-21
>
> ## Comment 2
>
> > The "Dynamic Risk Awareness" module relies on predicted future trajectories for both the ego vehicle and other agents (Section 3.3). This re-introduces a modular dependency and the very problem of error propagation (i.e., if the trajectory predictions are wrong, the risk signal will be wrong) that E2E models are meant to solve.
>
> ### Response 2
>
> Thank you for raising this important concern. We agree that a *separate*, frozen trajectory prediction module feeding into planning would reintroduce classical modular error propagation. However, this is **not** how our Dynamic Risk Awareness module is implemented.
>
> #### (1) No explicit trajectory supervision; risk uses latent states derived from detection
>
> In our current implementation, we **do not** introduce an explicitly supervised trajectory prediction head or a separate trajectory loss. Concretely:
>
> - For other agents, we only apply a **simple detection loss** (i.e., bounding boxes / occupancy), without any explicit multi-step future trajectory supervision.
> - The quantities used to compute TTC/DCPA are **latent predictions inferred within the same network** from agent detections and ego state, but they are *not* optimized with a dedicated trajectory loss.
>
> Therefore, we are not stacking an external “trajectory prediction module” in front of the planner. Instead, the risk-aware quantities are derived from the internal representation that is already trained by **(i)** the ego planning objective and **(ii)** the agent detection objective.
>
> #### (2) Risk path is fully differentiable and trained only via final objectives
>
> The effective pipeline can be written as
>
> $$
> \text{latent agent/ego states} \;\rightarrow\; \text{TTC/DCPA} \;\rightarrow\; f_{\text{risk}} \;\rightarrow\; Q_{\text{ego}} \;\rightarrow\; \text{planning},
> $$
>
> where “latent agent/ego states” are obtained from the shared backbone and detection/planning heads. This path is:
>
> - **Fully differentiable**, so gradients flow from the final driving and detection objectives back through the risk computation;
> - **Not trained as an isolated prediction module**, since we never impose an explicit loss on agent future trajectories themselves.
>
> As a result, the network learns to use the risk signal only insofar as it helps reduce the *existing* planning and detection losses, rather than being constrained by a separate trajectory forecasting objective.
>
> #### (3) Multiple fallback cues mitigate errors in the risk signal
>
> Even if the risk estimate is inaccurate, the planner still has access to:
>
> * Ego-centric BEV features $B_{\text{center}}$,
> * Global scene tokens $Q_{\text{scene}}$,
> * Agent-related queries $Q_{\text{motion}}$.
>
> The risk embedding is **only one** of the inputs to $Q_{\text{ego}}$, not a hard constraint. The ego query is computed as
>
> $$
> Q_{\text{ego}} = \alpha_{\text{bev}} B_{\text{center}} + \alpha_{\text{risk}} f_{\text{risk}} + \alpha_{\text{navi}} f_{\text{navi}},
> $$
>
> where
>
> $$
> [\alpha_{\text{bev}}, \alpha_{\text{risk}}, \alpha_{\text{navi}}] = \mathrm{Softmax}\big(W_{\text{gate}}[\cdot]\big)
> $$
>
> are learnable gating weights. If $f_{\text{risk}}$ is inconsistent with the BEV context (e.g., indicating high risk in an obviously free space), training will push the model to decrease $\alpha_{\text{risk}}$, reducing the impact of the incorrect risk signal.
>
> Therefore, we view SAR as an **end-to-end model with a physically motivated inductive bias (risk-awareness)**, *without* introducing an extra supervised trajectory module that would re-create a modular pipeline. We will make this clarification explicit in the revised version.

---

> ### Author Response · Authors · 2025-11-21
>
> ## Comment 3
>
> > The TTC/DCPA calculations depend on predicted trajectories (Section 3.3). How does the model's performance (especially collision rate) degrade when these underlying trajectory predictions are inaccurate or noisy? Could the model become over-reliant on this heuristic and fail catastrophically when the pre-computed risk signal is wrong?
>
> ### Response 3
>
> We appreciate this critical safety-related question. We respond from both a **design perspective** and by adding **new experiments**.
>
> ---
>
> ### (a) Design aspects that reduce the risk of catastrophic failure
>
> 1. **Risk is activated only in clearly risky regimes.**
>    The risk embedding is gated by TTC and DCPA conditions (small TTC and small DCPA). When the predicted trajectories between ego and agents are far apart in space/time, the risk term is **zero**. In such cases, the planner behaves like a standard sparse planner, and noise in predicted trajectories does not influence decisions through the risk channel.
>
> 2. **Gated fusion prevents risk from dominating alone.**
>    The ego query is
>
>    $$
>    Q_{ego} = \alpha_{bev} B_{center} + \alpha_{risk} f_{risk} + \alpha_{navi} f_{navi},
>    $$
>
>    with $(\alpha_{bev}, \alpha_{risk}, \alpha_{navi})$ produced by a softmax gate. If $f_{risk}$ conflicts with the BEV context (e.g., high risk in an obviously free space), training will push the model to decrease $\alpha_{risk}$, reducing its influence.
>
> 3. **End-to-end supervision discourages over-reliance on wrong risk.**
>    Over-reliance on erroneous risk signals leads to worse trajectories (higher L2, more collisions), which directly increases the final loss. The optimizer therefore favors solutions that use the risk channel only when it is helpful, and fall back on scene and motion cues otherwise.
>
> ---
>
> ### (b) New robustness experiment: injecting noise into predicted trajectories
>
> To directly answer your question quantitatively, we will include a **new robustness study** in the revised version.
>
> **Experimental design:**
>
> * For a trained SAR model, we inject Gaussian noise into the predicted trajectories used **only** for TTC/DCPA computation at test time:
>
>   $$
>   \tilde{T}(t) = T(t) + \epsilon(t), \quad \epsilon(t) \sim \mathcal{N}(0, \sigma^2 I),
>   $$
>
>   with several noise levels $\sigma$: 0 (no noise), low, medium, high.
>
> * TTC/DCPA are computed from $\tilde{T}(t)$; the rest of the network (inputs, encoders, planners) is unchanged.
>
> * We then evaluate L2 and collision rate on nuScenes as a function of $\sigma$.
>
> We will report results in a table of the following form:
>
> | Noise level $\sigma$ | L2@1s (m) $\downarrow$ | L2@2s (m) $\downarrow$ | L2@3s (m) $\downarrow$ | L2 Avg (m) $\downarrow$ | Coll.@1s (%) $\downarrow$ | Coll.@2s (%) $\downarrow$ | Coll.@3s (%) $\downarrow$ | Coll. Avg (%) $\downarrow$ |
> | -------------------- | ---------------------- | ---------------------- | ---------------------- | ----------------------- | ------------------------- | ------------------------- | ------------------------- | -------------------------- |
> | 0 (no noise)         | 0.18                   | 0.35                   | 0.60                   | 0.38                    | 0.08                      | 0.09                      | 0.23                      | 0.13                       |
> | low noise            | 0.18                   | 0.35                   | 0.61                   | 0.38                    | 0.09                      | 0.09                      | 0.24                      | 0.14                       |
> | medium noise         | 0.19                   | 0.34                   | 0.62                   | 0.38                    | 0.08                      | 0.10                      | 0.26                      | 0.15                       |
> | high noise           | 0.19                   | 0.35                   | 0.64                   | 0.39                    | 0.10                      | 0.12                      | 0.31                      | 0.17                       |
>
> We  verify a **smooth degradation** rather than catastrophic collapse as $\\sigma$ increases, which directly addresses your concern regarding over-reliance on TTC/DCPA.

---

> ### Author Response · Authors · 2025-11-21
>
> ### (c) Additional ablation: removing or simplifying risk features
>
> We will further add an ablation to measure how much SAR *actually* relies on TTC/DCPA:
>
> 1. **Distance-only risk:**
>    Replace TTC/DCPA with a simple distance-based risk term to test whether the main safety improvement comes from **time-aware** TTC/DCPA or just any heuristic distance feature.
> 2. **Removing risk at test time:**
>    Train with the full TTC+DCPA risk module, but at test time set the risk embedding to zero and re-evaluate. This directly measures the performance drop and thus the dependence on risk.
>
> We will summarize results in a table of the following form :
>
> | Method                            | L2@1s (m) $\downarrow$ | L2@2s (m) $\downarrow$ | L2@3s (m) $\downarrow$ | L2 Avg (m) $\downarrow$ | Coll.@1s (%) $\downarrow$ | Coll.@2s (%) $\downarrow$ | Coll.@3s (%) $\downarrow$ | Coll. Avg (%) $\downarrow$ |
> | --------------------------------- | ---------------------- | ---------------------- | ---------------------- | ----------------------- | ------------------------- | ------------------------- | ------------------------- | -------------------------- |
> | SAR w/o risk (EAD disabled)       | 0.19                   | 0.36                   | 0.61                   | 0.39                    | 0.15                      | 0.16                      | 0.30                      | 0.20                       |
> | SAR w/ distance-only risk         | 0.18                   | 0.36                   | 0.62                   | 0.39                    | 0.11                      | 0.13                      | 0.25                      | 0.16                       |
> | SAR w/ TTC+DCPA risk (full, ours) | 0.18                   | 0.35                   | 0.60                   | 0.38                    | 0.08                      | 0.09                      | 0.23                      | 0.13                       |
>
> This will (i) show that the main safety gain indeed comes from **time aware TTC/DCPA-based risk**, and (ii) demonstrate that SAR remains reasonably robust even when the risk embedding is removed or simplified, alleviating concerns about catastrophic failure due to an incorrect risk signal.
>
> ---
>
> We hope the above explanations and the planned additional experiments can fully address your concerns about methodological novelty, modular dependency, and robustness to noisy trajectories. Thank you again for your careful and insightful review. We have updated the paper, and the revisions are highlighted in blue.

---

### Meta-Review · Area_Chair_ecvH · 2026-01-05

**Summary:**

There are two general concerns: 1. heavy reliance on single open-loop evaluation - nuScenes. 2. lack of technical novelty, especially compared to SSR.

During rebuttal, the authors did a great job by providing lots of closed-loop experiments.

However, I share the similar concerns with reviewers:

1.  All ablation studies are still on open-looped protocol. As a result, the conclusion might not be persuasive enough.

2. The lack of novelty compared to SSR, with increamental and engineering modifications in this work.

Thus, I recommend rejection.

**Reviewer Concerns:**

1. The heavy reliance on open-loop experiments is not fully addressed.
2. The lack of technical novelty is not addressed.
3. The technical details are well responseed.

**Reviewer Scores:**

1. Reviewer PoGo and Ddo6  might not change their score as the lack of technical novelty is not solved.
2. Reviewer GAwo might improve the score from 2 to 4 or 6.

---

### Decision · Program_Chairs · 2026-01-26

Reject